# FUGAL:
# Feature-fortified Unrestricted Graph Alignment

**Aditya Bommakanti**
IIT Delhi
adityabommakanti2002@gmail.com

**Harshith Reddy Vonteri**
IIT Delhi
harshithreddyvonteri@gmail.com

**Konstantinos Skitsas**
Aarhus University
skitsas@cs.au.dk

**Sayan Ranu**
IIT Delhi
sayanranu@cse.iitd.ac.in

**Davide Mottin**
Aarhus University
davide@cs.au.dk

**Panagiotis Karras**
University of Copenhagen & Aarhus University
piekarras@gmail.com

## Abstract

The necessity to align two graphs, minimizing a structural distance metric, is prevalent in biology, chemistry, recommender systems, and social network analysis. Due to the problem's **NP**-hardness, prevailing graph alignment methods follow a *modular* and *mediated* approach, solving the problem restricted to the domain of intermediary graph representations or products like embeddings, spectra, and graph signals. Restricting the problem to this intermediate space may distort the original problem and are hence predisposed to miss high-quality solutions. In this paper, we propose an *unrestricted* method, FUGAL, which finds a permutation matrix that maps one graph to another by directly operating on their adjacency matrices with judicious constraint relaxation. Extensive experimentation demonstrates that FUGAL consistently surpasses state-of-the-art graph alignment methods in accuracy across all benchmark datasets without encumbering efficiency.

## 1 Introduction and Related Work

*Graph alignment* seeks to match a pair of graphs to each other, i.e., to correlate nodes of one graph to those of the other. For instance, biological systems such as protein-protein interaction networks and gene regulatory networks can be represented as graphs. The alignment of such biological networks across species reveals *orthologous* proteins or genes (i.e., homologous genes that evolved from a common ancestor) and thereby conveys the biological function of uncharted genes in one species through their better-studied counterparts in another species [37, 36]. The same problem also arises in other high-impact network science tasks [11], such as identifying users in social networks [22] and feature matching in computer vision [5, 6]. The problem can be formulated as an instance of the *quadratic assignment problem* (QAP) [13, 24] between nodes of the two graphs, which treats the edges in one graph as units of *flow* and the edges in the other graph as *distances* between nodes. This relation renders the problem **APX**-hard to approximate even within an approximation factor that grows linearly with the number of nodes [30, 13].

38th Conference on Neural Information Processing Systems (NeurIPS 2024).

## 1.1 Related Works

Owing to the problem's hardness, several heuristics have been proposed. Nevertheless, state-of-the-art graph alignment methods refrain from directly addressing the edge-aware QAP. Instead, they craft intermediate representations of nodes that allow for the computation of *similarities* and settle for solving an *assignment problem* over those representations. We call these methods *mediated* due to their restriction to *intermediary* graph representations. While the transformation from the original graph space to an intermediate space enables computational efficiency, the transformation incurs loss of information. In this work, we propose an *unrestricted* graph alignment method that avoids restricting the problem to an intermediate space, while also retaining efficiency. We call our method "unrestricted" rather than "unmediated" since, while we retain the full graph information in the core QAP, we also avail of help from mediated representations to solve the QAP. In the subsequent discussion, we summarize the various mediated and unmediated approaches in the literature.

**Mediated Approaches:** GWL [44] jointly learns embeddings and alignments using the dissimilarity notion of Gromov-Wasserstein discrepancy; it estimates distance matrices using the embeddings when learning the optimal transport, and regularizes the learning of embeddings using the learned transport. S-GWL [43] addresses the scalability drawback of GWL by adopting a partitioning method on the input graphs. CONE [4] models intra-network proximity with node embeddings and uses them to match nodes across networks after aligning embedding subspaces. REGAL [17] identifies node matchings by greedily aligning their latent feature representations learnt from graph structures. GRAMPA [13] constructs a similarity matrix as a weighted sum of outer products between all pairs of eigenvectors of the two graphs. GRASP [20] uses the spectral properties of the graphs grounded on the eigenvectors of their normalized Laplacian matrices. IsoRank [37] uses neighborhood similarity to extract structural graph information and recursively updates the score of a node pair using the score of their neighbors. GRAAL [25] is a greedy alignment method that matches nodes using a similarity score based on a dictionary of small frequent graph patterns. GOT [31] employs the probabilistic distribution of smooth graph signals defined with respect to the graph topology, and seeks alignments by minimizing the distance between these graph signal distributions. fGOT [32] adopts a dissimilarity metric that aligns two graphs using the probability distribution of data generated via graph filters. PARROT [46] presents a semi-supervised methodology which encodes graph topology through random walks with restart (RWR) for a position-aware transport cost and addresses a regularized Optimal Transport (OT) problem to determine node mappings. GW [35] and FGW [40] compute Gromov-Wasserstein discrepancy using similarity matrices of shortest path distances between nodes.

**Unmediated Approaches:** FAQ [42] is an unmediated algorithm that addresses the QAP by relaxing constraints to attain computational tractability. GLAG [14] proposes a problem formulation that retains the full graph information and relaxes the permutation constraints. As we will see in § 5, these methods of relaxing constraints lead to inferior accuracy.

**Unrestricted Approaches:** In addition to the full graph information, PATH [45] and FGM [49] also use feature matching, while DSPP [10] employs all-pairs-shortest-paths for graph alignment. We characterize these methods as unrestricted, since they also avail of help from mediated representations. As we will see in § 5, these methods are significantly inferior to FUGAL in terms of accuracy.

## 1.2 Contributions

**Optimization problem formulation:** We present FUGAL (*Feature-fortified Unrestricted Graph Alignment*), a graph alignment method that retains full graph information by integrating the quadratic assignment problem (QAP) in the optimization objective. To augment quality, we utilize a regularizer in the form of a linear assignment (LAP) supplement incorporating graph structural features.
**Unrestricted solution:** FUGAL relaxes the solution space to *doubly stochastic* matrices and uses a customized optimization strategy that guides the Frank-Wolfe algorithm [16] through a Sinkhorn distance objective [7] to steer the resulting doubly stochastic solution towards a *quasi-permutation* matrix. We call our approach *unrestricted*, as it does not rely *solely* on intermediate representations. On the other hand, it is not entirely *unmediated*, as the LAP regularizer using structural features is mediated. Thereby, we retain the full graph information and also enable mediating representations to efficiently guide the optimization process and thereby enable both efficacy and tractability.
**Experimental evaluation:** Through extensive experimentation with real-world and synthetic datasets across varying graph density and noise levels, we demonstrate that FUGAL outperforms state-of-the-art graph alignment methods in accuracy without a detrimental efficiency overhead.

## 2 Problem Formulation

**Definition 2.1.** Let $\mathcal{G}(\mathcal{V}, \mathcal{E})$ denote an unlabelled, undirected *graph*, where $\mathcal{V}$ is the set of nodes each identified by a number $[n] = \{1, \dots, n\}$ and $\mathcal{E} \subseteq \mathcal{V} \times \mathcal{V}$ is the edge set. The adjacency matrix of $\mathcal{G}$ is $\mathbf{A} \in \{0,1\}^{n \times n}$ such that $a_{ij} = a_{ji} = 1$ if and only if $(i,j) \in \mathcal{E}$.

We denote an all-ones vector as $\mathbf{1}$, an all-ones square matrix as $\mathbf{J}$, and an all-zero square matrix as $\mathbf{O}$. The dimensions of entities are inferred from the equations employing them.

**Definition 2.2.** We denote the set of binary-valued *permutation matrices* as $\mathbb{P}^n = \{\mathbf{P} \in \{0,1\}^{n \times n} : \mathbf{P}\mathbf{1} = \mathbf{1}, \mathbf{P}^\top \mathbf{1} = \mathbf{1}\}$ and that of real-valued *doubly stochastic* matrices as $\mathbb{W}^n = \{\mathbf{W} \in [0,1]^{n \times n} : \mathbf{W}\mathbf{1} = \mathbf{1}, \mathbf{W}^\top \mathbf{1} = \mathbf{1}\}$.

**Definition 2.3.** Let $\mathbf{A} = [a_{ij}]_{i \in [n], j \in [m]} \in \mathbb{R}^{n \times m}$. We denote the *Frobenius norm* as the entry-wise 2-norm $\|\mathbf{A}\|_F = \left( \sum_{i=1}^{n} \sum_{j=1}^{m} |a_{ij}|^2 \right)^{1/2}$.

**Definition 2.4.** We denote the *trace* of a matrix $\mathbf{A}$ as $\mathrm{tr}(\mathbf{A})$.

**Theorem 2.5.** *A doubly-stochastic matrix $\mathbf{A}$ with $\mathrm{tr}(\mathbf{A}^\top (\mathbf{J} - \mathbf{A})) = 0$ is a permutation matrix.*

*Proof.* From $\mathrm{tr}(\mathbf{A}^\top (\mathbf{J} - \mathbf{A})) = 0$ follows that $\sum_i \sum_j a_{ij} \cdot (1 - a_{ij}) = 0$. Since $\mathbf{A}$ is doubly-stochastic, $0 \leq a_{ij} \leq 1$ for all $i$ and $j$. Thus, $a_{ij} \cdot (1 - a_{ij}) \geq 0$ for $1 \leq i, j \leq n$. Therefore, $a_{ij} \cdot (1 - a_{ij}) = 0$ for all $i$ and $j$. As a consequence, $a_{ij} \in \{0, 1\}$ for each $i$ and $j$. Given that $\mathbf{A}$ is doubly-stochastic and all its entries are either 0 or 1, by Definition 2.2, $\mathbf{A}$ is a permutation matrix. $\square$

**Problem 1** (Unmediated Graph Alignment). *Consider two graphs $\mathcal{G}_1 := (\mathcal{V}_1, \mathcal{E}_1)$ and $\mathcal{G}_2 := (\mathcal{V}_2, \mathcal{E}_2)$ with adjacency matrices $\mathbf{A}, \mathbf{B}$ respectively. The objective of* unmediated graph alignment *is to identify a bijection $f : \mathcal{V}_1 \to \mathcal{V}_2$ between the two graphs that minimizes the number of edge disagreements. Formally, the problem is expressed as:*

$$\min_{\mathbf{P} \in \mathbb{P}^n} \|\mathbf{A}\mathbf{P} - \mathbf{P}\mathbf{B}\|_F^2, \tag{1}$$

*where $\mathbb{P}^n$ denotes the set of permutation matrices.*

The appellation *unmediated* denotes that we seek a correspondence among nodes without using any information other than the adjacency matrix. The problem is an instance of the **NP**-hard *quadratic assignment problem* (QAP) [24].

Due to the problem's hardness, a popular approximation path utilizes *intermediaries* such as node embeddings. A *mediated* graph alignment is thus expressed as a linear assignment between embeddings rather than a quadratic assignment between adjacency matrices:

$$\min_{\mathbf{P} \in \mathbb{P}^n} \|\mathbf{E}_1 - \mathbf{P}\mathbf{E}_2\|_F^2 \tag{2}$$

where $\mathbf{E}_k \in \mathbb{R}^{|\mathcal{V}_k| \times F}$ is the embedding matrix of $\mathcal{G}_k$ and $\mathbf{E}_k[i, :]$ is the $F$-dimensional vector representation of node $i$ of $\mathcal{G}_k$. The optimization problem in Equation (2) is a *linear assignment problem* (LAP), which is solvable optimally in $\mathcal{O}(N^3)$ by the Hungarian algorithm [26], while sub-optimal solutions reduce complexity to $\mathcal{O}(N^2)$.

**Extension to graphs of unequal sizes.** Consider two graphs $\mathcal{G}_1$ and $\mathcal{G}_2$ with node counts $n_1$ and $n_2$, respectively ($n_1 < n_2$). To enable alignment despite the size difference, we augment $\mathcal{G}_1$ with $(n_2 - n_1)$ isolated dummy nodes and discard mappings involving dummy nodes from the output.

## 3 FUGAL

To design FUGAL, we augment the core QAP of Eq. (1) with a LAP supplement that leverages simple *structural* graph features (§ 3.1) to form a unified optimization problem over the set of permutation matrices $\mathbb{P}^n$ (§ 3.2). As this problem is **NP**-hard, we relax its solution space to the set of *doubly stochastic* matrices $\mathbb{W}^n$ (§ 3.3), a superset of the set of permutation matrices. We refine the solution to obtain a *quasi-permutation matrix*, i.e., *almost* a permutation matrix, which we adjust to a permutation matrix that signifies a valid alignment by solving a simple LAP using the *Hungarian* algorithm [26]. We dub this approach "unrestricted" as it eschews the information loss incurred by mediated solutions, which rely solely on intermediary representations. However, we still employ supplementary mediating representations to ensure tractability and efficiency.

## 3.1 LAP Formulation

Here, we formalize the Linear Assignment Problem (LAP), which is auxiliary to our framework. We construct a node feature vector using four structural features proposed in NETSIMILE [2]. This includes (1) $d_i$, the degree of node $v_i$, (2) $c_i$, the clustering coefficient of $v_i$, (3) $\bar{d}_{N_i}$, the mean degree of $v_i$'s neighbors, (4) and $\bar{c}_{N_i}$, the mean clustering coefficient of $v_i$'s neighbours. Other features, such as betweenness centrality, PageRank, may also be used. Ultimately, the decision resides on the trade-off between the utility of including these features on alignment quality and the efficiency of computing these features.

Using these features, we construct a feature matrix $\mathbf{F}_k \in \mathbb{R}^{|\mathcal{V}_k| \times 4}$ for each graph $\mathcal{G}_k$ and, by the rationale that the structural features of corresponding nodes are similar, we formulate a Linear Assignment Problem for $\mathcal{G}_1$ and $\mathcal{G}_2$ as:

$$\min_{\mathbf{P} \in \mathbb{P}^n} \|\mathbf{F}_1 - \mathbf{P}\mathbf{F}_2\|_F^2 \tag{3}$$

By the Frobenius norm definition, Eq. (3) is equivalent to:

$$\min_{\mathbf{P} \in \mathbb{P}^n} \sum_i \|\mathbf{F}_1[i,:] - \sum_j \mathbf{P}_{ij}\mathbf{F}_2[j,:]\|_F^2 \tag{4}$$

Utilizing the property of permutation matrices that each row contains only one 1, we reformulate Eq. (4) to:

$$\min_{\mathbf{P} \in \mathbb{P}^n} \sum_{i,j} \mathbf{P}_{ij}\|\mathbf{F}_1[i,:] - \mathbf{F}_2[j,:]\|_F^2 = \min_{\mathbf{P} \in \mathbb{P}^n} \sum_{i,j} \mathbf{P}_{ij}\mathbf{D}_{ij} \tag{5}$$

where $\mathbf{D}$ is a distance matrix with $\mathbf{D}_{ij}$ denoting the squared Euclidean distance between $\mathbf{F}_1[i,:]$ and $\mathbf{F}_2[j,:]$. Since each row $\mathbf{P}[i,:]$ contributes exactly one term to this sum, being the element of $\mathbf{D}$ corresponding to the single 1 entry in $\mathbf{P}[i,:]$, the result is equal to the trace of the matrix product:

$$\min_{\mathbf{P} \in \mathbb{P}^n} \operatorname{tr}(\mathbf{P}^\top \mathbf{D}) \tag{6}$$

## 3.2 Optimization Problem

Our problem formulation augments the QAP of Eq. (1) with a LAP regularizing term as in Eq. (6):

$$\min_{\mathbf{P} \in \mathbb{P}^n} \|\mathbf{AP} - \mathbf{PB}\|_F^2 + \mu \cdot \operatorname{tr}(\mathbf{P}^\top \mathbf{D}) \tag{7}$$

where $\mathbf{A}$ and $\mathbf{B}$ denote the adjacency matrices of $\mathcal{G}_1$ and $\mathcal{G}_2$, respectively, $\mathbf{D}$ follows Eq. (6), and $\mu$ regulates the LAP's significance; since $\mathbf{P}\mathbf{P}^\top = \mathbf{I}$, this is expanded to:

$$\min_{\mathbf{P} \in \mathbb{P}^n} \operatorname{tr}(\mathbf{A}^\top \mathbf{A}) + \operatorname{tr}(\mathbf{B}^\top \mathbf{B}) - 2\operatorname{tr}(\mathbf{APB}^\top \mathbf{P}^\top) + \mu \cdot \operatorname{tr}(\mathbf{P}^\top \mathbf{D}) \tag{8}$$

equivalently, ignoring constant terms and reversing the sign,

$$\max_{\mathbf{P} \in \mathbb{P}^n} \operatorname{tr}(\mathbf{APB}^\top \mathbf{P}^\top) - \mu \cdot \operatorname{tr}(\mathbf{P}^\top \mathbf{D}) \tag{9}$$

In the case of $\mu = 0$, the first term alone corresponds to the maxQAP problem [30], which is **APX**-hard to approximate even within an approximation factor that grows linearly with the number of nodes. Given this hardness of the QAP alone and the fact that relaxing combinatorial constraints often results in a substantial deterioration of solution quality, we introduce the LAP regularization to ground the QAP solution on pragmatic features and thereby guide it, even after we relax combinatorial constraints.

## 3.3 Approximating the Optimization Problem

The problem in Eq. (9) is **NP**-hard, due to the non-convex nature of the space of permutation matrices [24]. A natural way to overcome this hardness is to enlarge the allowed solution space to the convex set of *doubly stochastic* matrices $\mathbb{W}^n$, as considered in FAQ [42]:

$$\min_{\mathbf{P} \in \mathbb{W}^n} -\operatorname{tr}(\mathbf{APB}^\top \mathbf{P}^\top) + \mu \cdot \operatorname{tr}(\mathbf{P}^\top \mathbf{D}) \tag{10}$$

Since the problem in Eq. (10) calls to minimize a function subject to linear constraints implied by $\mathbf{P} \in \mathbb{W}^n$, the solution can be efficiently found [3] by algorithms such as *Adam* [23] and *Frank-Wolfe* [16]. The FAQ algorithm [42] follows such an approach to solve the relaxed optimization with the Frank-Wolfe algorithm and project the solution back onto $\mathbb{P}^n$, yet addresses exclusively the first, QAP term in Eq. (10). To further augment quality, as we elaborate later in Section 5, we include the LAP term in Eq. (10) and also add a regularizing term that guides the solution towards a quasi-permutation matrix. By Theorem 2.5, which establishes that a doubly-stochastic matrix $\mathbf{P}$ with $\mathrm{tr}(\mathbf{P}^\top(\mathbf{J} - \mathbf{P})) = 0$ is a permutation matrix, we rewrite the problem in Eq. (9) as:

---

**Algorithm 1** FINDQUASIPERMUTATION $(\mathbf{A}, \mathbf{B}, \mathbf{D}, \mu, T)$

---

**Input:** Adjacency Matrices $\mathbf{A}$, $\mathbf{B}$, Distance matrix $\mathbf{D}$, control parameter $\mu$, num iters $T$
**Output:** Quasi-Permutation matrix $\mathbf{Q}$
**Notation:**
    $f(\mathbf{P}) : -\mathrm{tr}(\mathbf{APB}^\top\mathbf{P}^\top) + \mu \cdot \mathrm{tr}(\mathbf{P}^\top\mathbf{D})$
    $g(\mathbf{P}) : \mathrm{tr}(\mathbf{P}^\top(\mathbf{J} - \mathbf{P}))$
1:  $\mathbf{Q} \leftarrow \mathbf{1} \cdot \mathbf{1}^\top / n$
2:  **for** $\lambda = 0$ **to** $T - 1$ **do**
3:    **for** $it = 1$ **to** 10 **do**
4:      $grad \leftarrow \nabla f(\mathbf{Q}) + \lambda \cdot \nabla g(\mathbf{Q})$
5:      $q_{it} \leftarrow \arg\min_{q \in \mathbb{W}^n} \langle grad, q \rangle$ \\ Sinkhorn-Knopp
6:      $\alpha \leftarrow \frac{2}{2+it}$
7:      $\mathbf{Q} \leftarrow \mathbf{Q} + \alpha \cdot (q_{it} - \mathbf{Q})$
8:    **end for**
9:  **end for**
10: **return** $\mathbf{Q}$

---

$$\min_{\mathbf{P} \in \mathbb{W}^n} -\mathrm{tr}(\mathbf{APB}^\top\mathbf{P}^\top) + \mu \cdot \mathrm{tr}(\mathbf{P}^\top\mathbf{D})$$
$$\text{Constraints: } \mathrm{tr}(\mathbf{P}^\top(\mathbf{J} - \mathbf{P})) = 0 \tag{11}$$

and turn the constraint to a regularizer with parameter $\lambda$:

$$\min_{\mathbf{P} \in \mathbb{W}^n} -\mathrm{tr}(\mathbf{APB}^\top\mathbf{P}^\top) + \mu \cdot \mathrm{tr}(\mathbf{P}^\top\mathbf{D}) + \lambda \cdot (\mathrm{tr}(\mathbf{P}^\top(\mathbf{J} - \mathbf{P}))) \tag{12}$$

Equivalently, by reformulating the constraints:

$$\min_{\mathbf{P}} -\mathrm{tr}(\mathbf{APB}^\top\mathbf{P}^\top) + \mu \cdot \mathrm{tr}(\mathbf{P}^\top\mathbf{D}) + \lambda \cdot (\mathrm{tr}(\mathbf{P}^\top(\mathbf{J} - \mathbf{P})))$$
$$\text{Constraints: } \mathbf{P1} = \mathbf{1}, \mathbf{P}^\top\mathbf{1} = \mathbf{1}, 0 \leq \mathbf{P}_{ij} \leq 1 \tag{13}$$

We solve the problem in eq. (13) for $\lambda = 0$ by the Frank-Wolfe (FW) algorithm [16] with updates guided by an objective computed via the Sinkhorn-Knopp algorithm [7], due to the computational efficiency they confer. We use the solution to this optimization problem as a warm start, and refine it by gradually increasing $\lambda$ over $T$ iterations, each initiating with the solution obtained in the preceding one and solving the problem in Eq. (13) by FW. Alg. 1 outlines the process.

**Rounding Algorithm:** Alg. 1 yields a *quasi-permutation* matrix $\mathbf{Q}$. Next, to obtain an one-to-one mapping between nodes of $\mathcal{G}_1$ and $\mathcal{G}2$, we need to adjust $\mathbf{Q}$ to a permutation matrix by rounding. We pose this problem as an assignment problem, maximizing the sum of $\mathbf{Q}_{ij}$ entries selected for rounding up to 1, while rounding the rest down to 0, and solve it optimally by the Hungarian algorithm [26]. Alg. 2 in the appendix presents the complete FUGAL pseudocode.

## 4   Customized Optimization Strategy for Node Alignment

In this section, we elucidate the intricacies of Algorithm 1, which derives a quasi-permutation matrix, focusing on two pivotal steps: *(i)* initialization of the quasi-permutation matrix; *(ii)* finding the local solution for a given $\lambda$.

**Initialization:** Any doubly stochastic matrix is a viable option for initialization. However, we opt for an *uninformative flat* matrix, $\mathbf{1} \cdot \mathbf{1}^\top / n$. Our empirical observations indicated that this initialization consistently performs well across diverse datasets, contrary to *informative* initializations like the *identity matrix*, which exhibit inconsistency in performance, as we further elaborate in Section A.8.

**Local Solution for a given $\lambda$:** Given a specific $\lambda$, our objective is to solve the optimization problem of Eq. (13) under linear constraints. To achieve this, we employ the Frank-Wolfe algorithm (FW), a successive first-order optimization technique devised for solving convex quadratic programs [16]. While FW is a widely utilized solver as a subroutine for QAP algorithms, we tailor its application to FUGAL. Specifically, each iteration commences from the local solution obtained in the previous iteration and involves the following steps:
**Computing the Gradient:** The gradient of the objective function $f(\mathbf{P}) = -\mathrm{tr}(\mathbf{APB}^\top\mathbf{P}^\top) + \mu \cdot \mathrm{tr}(\mathbf{P}^\top\mathbf{D})$ with respect to $\mathbf{P}$, evaluated at $\mathbf{Q}$, is $\nabla f(\mathbf{Q}) = -\mathbf{AQB}^\top - \mathbf{A}^\top\mathbf{QB} + \mu \cdot \mathbf{D}$. Additionally,

the gradient of the constraint function $g(\mathbf{P}) = \text{tr}(\mathbf{P}^\top(\mathbf{J} - \mathbf{P}))$ with respect to $\mathbf{P}$, evaluated at $\mathbf{Q}$, is $\nabla g(\mathbf{Q}) = \mathbf{J} - 2\mathbf{Q}$.

**Updating Q:** A critical step involves determining the doubly-stochastic matrix $q_{it}$ that minimizes the inner product $\langle grad, q \rangle$, where $grad$ is the current gradient. Prior work [42] applies the Hungarian algorithm to obtain a permutation matrix to that end, which, however, may not yield the optimal answer and incurs $\mathcal{O}(n^3)$ cost. Contrarily, we obtain a proper doubly stochastic matrix to that end.

**Definition 4.1** (Optimal Transport Distance Between $r$ and $c$). Given a $n \times n$ cost matrix $\mathbf{M}$, the cost of mapping an $n$-dimensional probability vector $r$ to $c$, using a transportation matrix (or joint probability) $\mathbf{P}$ is quantified as $\langle \mathbf{P}, \mathbf{M} \rangle$. The following problem:

$$\min_{\mathbf{P} \in U(r,c)} \langle \mathbf{P}, \mathbf{M} \rangle. \tag{14}$$

is an *optimal transport* problem between $r$ and $c$ given cost $\mathbf{M}$, where

$$U(r,c) = \{\mathbf{P} \in \mathbb{R}_+^{n \times n} \mid \mathbf{P}\mathbf{1} = r, \mathbf{P}^\top \mathbf{1} = c\} \tag{15}$$

To render this optimal transport objective *strictly* convex and thus efficiently solvable by the *matrix scaling* Sinkhorn-Knopp fixed-point iteration algorithm [38] via matrix-vector products, we regularize it with an entropic penalty $h(\mathbf{P})$ that yields the *Sinkhorn distance* objective [7]:

$$\min_{\mathbf{P} \in U(r,c)} \langle \mathbf{P}, \mathbf{M} \rangle - \frac{1}{\kappa} h(\mathbf{P}) \tag{16}$$

where $h(\mathbf{P}) = -\sum_{i,j=1}^n \mathbf{P}_{ij} \log \mathbf{P}_{ij}$, $\kappa \in (0, \infty]$, which becomes equivalent to the transport distance for suitably large $\kappa$ [7]. The method exhibits excellent performance in practice with $\mathcal{O}(n^2)$ empirical time complexity.

Setting $r$ and $c$ to $\mathbf{1}$ (the all-ones vector) in Eq. (15), $U(r,c)$ becomes the space of doubly stochastic matrices, hence Eq. (14) with $\mathbf{M} = grad$ captures our update step objective. Thus, we find the doubly stochastic matrix $q_{it}$ that minimizes $\langle grad, q \rangle$ by the Sinkhorn-Knopp algorithm and update $\mathbf{Q}$ as $\mathbf{Q} \leftarrow \mathbf{Q} + \alpha \cdot (q_{it} - \mathbf{Q})$, with the step size $\alpha$ following the conventional choice $\alpha = 2/(2 + it)$.

**Complexity Analysis:** The complexity of FUGAL is $\mathcal{O}(n^3)$, which is in line with the majority of the baselines. A detailed derivation and comparison of FUGAL's complexity with baselines is provided in App. A.1.The $\mathcal{O}(n^3)$ complexity stems from the need to perform matrix multiplications, a core operation in FUGAL as well as the baselines. A thorough evaluation of empirical running times (§ 5.5) also demonstrates the practical scalability of FUGAL.

## 5 Experiments

In this section, we present a comprehensive evaluation of FUGAL vs. state-of-the-art graph alignment baselines on real and synthetic data sets with varying noise levels.

### 5.1 Datasets

**Real Graphs.** Table 1 summarizes the real-world datasets used to benchmark FUGAL.The last three data sets in the table are evolving graphs mandating challenging ground-truth alignments.

**Synthetic Graphs.** We employ Newmann-Watts (NW) [21] graphs, characterized by small-world properties and a high clustering coefficient. We generate NW graphs with 1000 nodes, number of neighbors per node $k = 7$, and a rewiring probability of $p = 0.1$. For each graph, we generate 5 noisy variants, perform alignments on each, and report average results. Given the obtained alignment set $\mathcal{P}$ and the ground truth set of alignments $\mathcal{P}_{real}$, we calculate *accuracy* as $\frac{|\mathcal{P} \cap \mathcal{P}_{real}|}{|\mathcal{P}|} \cdot 100$.

**Noise Types.** As in prior work [4, 20, 43], we introduce perturbations to the adjacency matrix by either removing or adding edges. We employ two noise types: *one-way* noise removes edges from the target graph, while *bimodal* noise removes and restores the same number of edges.

Table 1: Real-graph nodes $n$, edges $m$, and network type.

| Dataset | $n$ | $m$ | Type |
|---|---|---|---|
| Arenas [27] | 1 133 | 5 451 | communication |
| inf-euroroad [1] | 1 174 | 1 417 | infrastructure |
| bio-celegans [9] | 453 | 2 025 | biological |
| ca-netscience [33] | 379 | 914 | collaboration |
| ACM [48] | 9 872 | 39 561 | citation |
| DBLP [48] | 9 916 | 44 808 | citation |
| MultiMagna [41] | 1 004 | 8 323 | biological |
| HighSchool [15] | 327 | 5 818 | proximity |
| Voles [8] | 712 | 2 391 | proximity |

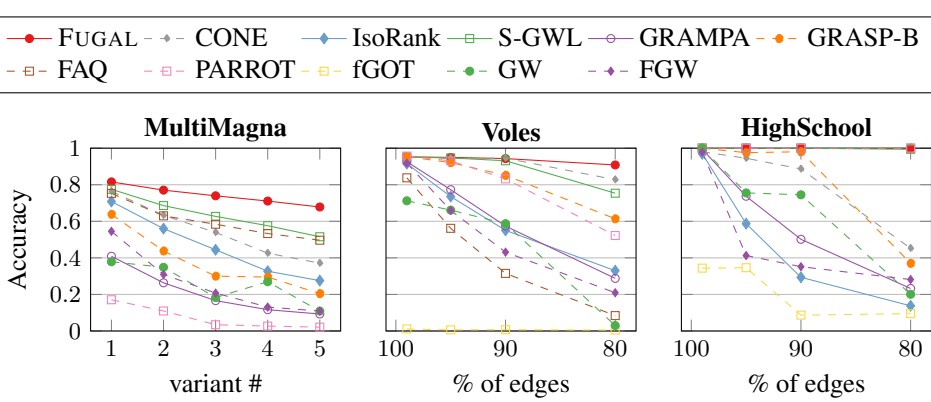

Figure 1: Accuracy, real graphs with real noise.

## 5.2 Experimental Setup

We ran all experiments on a 40-core Intel Xeon E5-2687W CPU machine @3.10GHz with Python implementations of FUGAL[1] and competitors;[2] the latter include CONE [4], IsoRank [37], S-GWL [43], GRAMPA [13], GRASP-B [19], FAQ [42], PARROT [46], fGOT [32], GOT [31], GW [35] and FGW [40]. Due to scalability limitations, we excluded fGOT from consideration for graphs with more than 1000 nodes, on which it failed to terminate within 5 hours. Moreover, due to the inability of GOT, PATH, and DSPP to scale for the smallest dataset in our analysis, we assess them separately on smaller graphs in Appendix A.4. We exclude GWL from evaluation in favor of its scalable and superior variant, S-GWL [39]. We omit from the comparison algorithms such as GRAAL [25], GLAG [14] and REGAL [17] due to their inferior performance [39, 29]. As we focus on non-attributed graphs, we exclude FINAL [47], which is equivalent to IsoRank on graphs without attributes. For the prerequisite similarity score in IsoRank, we devise a customized weight scheme as $sim(u,v) = 1 - \frac{|d_u - d_v|}{\max\{d_u, d_v\}}$, where $d_u = |N(u)|$ denotes the degree of node $u$. With all baselines, we use author-recommended parameters and derive node matchings from similarity scores using the Hungarian algorithm. In Appendix A.5, we benchmark FUGAL against S-GWL and CONE in terms of Matched Neighborhood consistency (MNC) [4] and the Frobenius norm between aligned graph adjacency matrices.

## 5.3 Accuracy on varying noise

**Graphs with real Noise:** We evaluate all algorithms on accuracy with three real-world networks: MultiMagna, Voles, and High School. MultiMagna represents a yeast protein-protein interaction (PPI) network and noisy variants incorporating an additional $q\%$ of low-confidence interactions, with $q \in \{5, 10, 15, 20, 25\}$. High School and Voles are temporal proximity networks; we align the last graph version to versions containing 80%, 85%, 90%, and 99% of edges. Figure 1 presents our results. FUGAL consistently achieves accuracy surpassing its counterparts across all datasets, with S-GWL being the closest baseline on average. On MultiMagna, FUGAL attains a 4% improvement over the next best algorithm, S-GWL, on the first graph variant, and this gap steadily increases to 16% on the last variant. On Voles, CONE and S-GWL follow FUGAL's accuracy with up to 90% of edges, yet with 80% of edges, they achieve 83% and 75% accuracy, respectively, vs. 90% of FUGAL. On the High School network, FUGAL, FAQ and PARROT align graphs perfectly, while S-GWL attains near-perfect alignment. IsoRank, GRAMPA, GRASP-B fGOT, GW and FGW fall short of FUGAL's performance across all three datasets. Despite performing comparably to FUGAL on the HighSchool dataset, FAQ and PARROT exhibit notably poorer performance on other datasets. The consistently superior performance of FUGAL underscores its robustness.

**Large Real Graphs with Partially Aligned nodes:** ACM and DBLP are two co-authorship networks of the ACM Digital Library and DBLP bibliography. In these networks, nodes represent authors, and an edge exists between two authors if they have collaborated on at least one publication. Across both networks, there are 6,325 authors who appear in both. Although both networks are attributed, we did not incorporate this information in our experiments. Note that S-GWL and GRASP-B are not scalable for networks of this magnitude, hence omitted from the analysis. Furthermore, the experiment was conducted in an unsupervised manner, meaning that the methods were not provided with any prior information regarding node alignment. Our results, detailed in Table 2, showcase

---

[1]Code and data at `https://github.com/idea-iitd/Fugal`.
[2]Source code from `https://github.com/constantinosskitsas/Framework_GraphAlignment`.

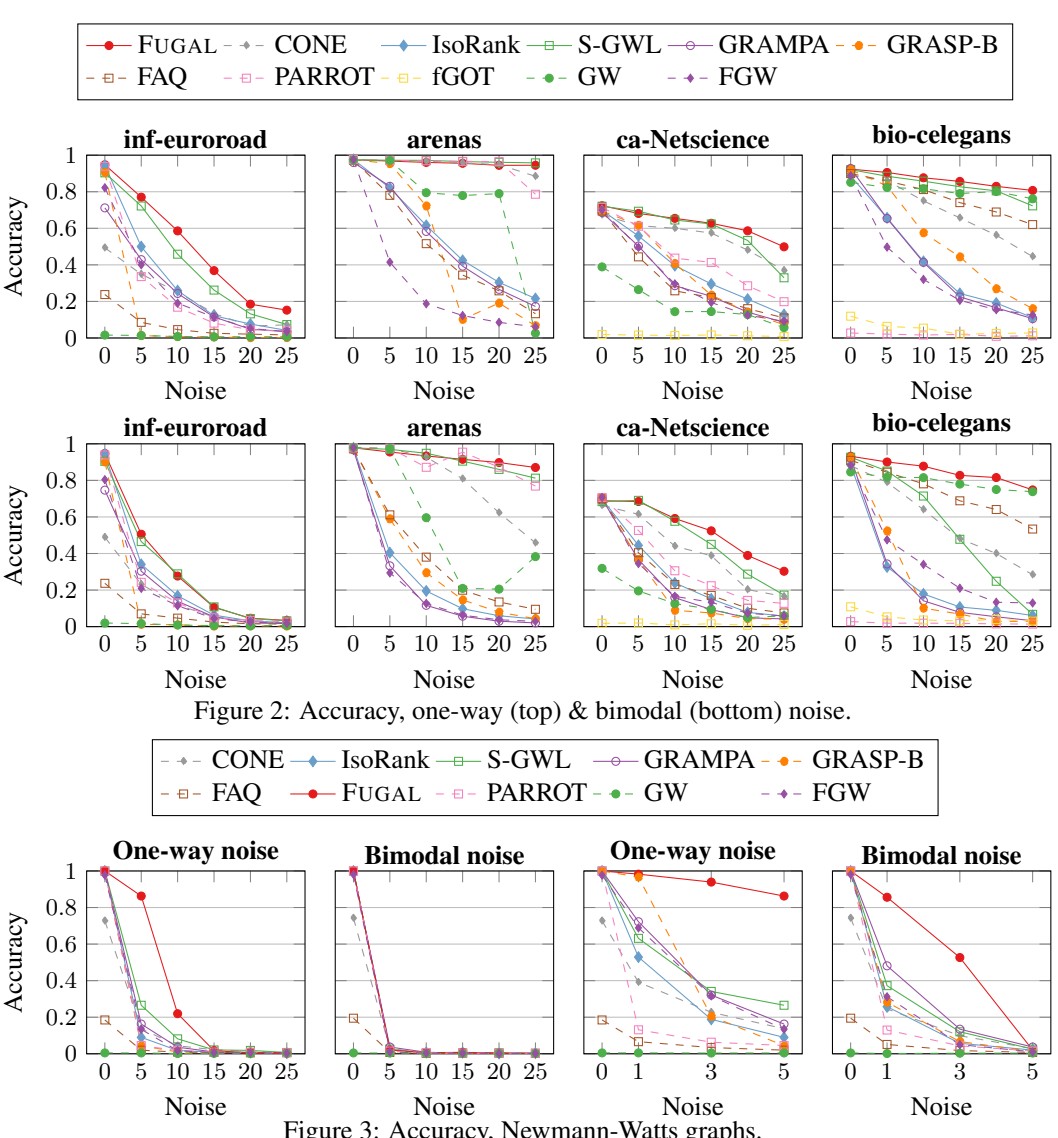

Figure 2: Accuracy, one-way (top) & bimodal (bottom) noise.

Figure 3: Accuracy, Newmann-Watts graphs.

the fraction of correctly aligned nodes out of the 6,325 aligned nodes. FUGAL demonstrated a significant improvement of 30% compared to the closest baseline, CONE. This underscores the superior scalability of FUGAL without compromising accuracy.

Table 2: Accuracy in alignment across ACM-DBLP.

|  | CONE | IsoRank | GRAMPA | FAQ | PARROT | GW | FGW | FUGAL |
|---|---|---|---|---|---|---|---|---|
| **Accuracy** | 0.183 | 0.042 | 0.011 | 0.025 | 0.000 | 0.028 | 0.012 | **0.487** |

**Real Graphs with Injected Noise:** Figure 2 illustrates the results on real datasets subject to synthetic one-way and bimodal noise. Consistent with the trends observed in real noise, FUGAL exhibits superior performance across all evaluated networks and noise types. This consistent superior performance of FUGAL establishes it as a robust graph alignment solution. Appendix A.3 zooms in on the performance of FUGAL vs. baselines with noise levels in the range 0% to 5% to better highlight the performance superiority of FUGAL.

**Synthetic Graphs:** Figure 3 portrays accuracy results on Newmann-Watts graphs of 1000 nodes with node degree $k = 7$ and rewiring probability $p = 0.1$ subject to synthetic noise. Under one-way noise, all methods except CONE and FAQ achieve perfect alignment at 0% noise. With noise of 5% and 10%, FUGAL attains a 60% and 14% gain, respectively, over the 2nd-best method, S-GWL. Beyond these noise levels, all methods experience failures. Bimodal noise at 0% results in perfect alignment for most methods. However, alignment failures occur as noise grows. Figure 3 further zooms in noise levels in the range of 0% to 5%. FUGAL significantly outperforms all baselines under one-way noise, achieving a margin of 60% at 3% and 5% levels. Moreover, FUGAL performs

superiorly in the bimodal noise within the 5% noise threshold, gaining nearly 40% at 1% and 3% noise levels. These results underscore the efficacy of FUGAL in handling diverse graph structures.

## 5.4 Varying Density

Here, we examine performance under varying graph density. In Newmann-Watts graphs, the rewiring probability parameter $p$ affects the edge density of sampled graphs for a fixed number of nodes $n$, while the parameter $k$, representing the number of nearest neighbors per node, affects the minimum and expected degree. Figure 4 shows our results when varying $p$ and $k$ in NW graphs comprising 1000 nodes. Methods other than FUGAL consistently fail to handle sparse graphs (low $p$). However, FUGAL attains accuracy 92% at $p = 0.25$, outperforming S-GWL, which achieves only 54%. Sparse graphs pose a challenge for alignment, as they provide less discriminating evidence in terms of density differentials. When varying $k$, FUGAL consistently achieves near-perfect alignment, surpassing all baselines. These findings corroborate the resilience of FUGAL across graph densities and its adaptability to varying degrees of connectivity.

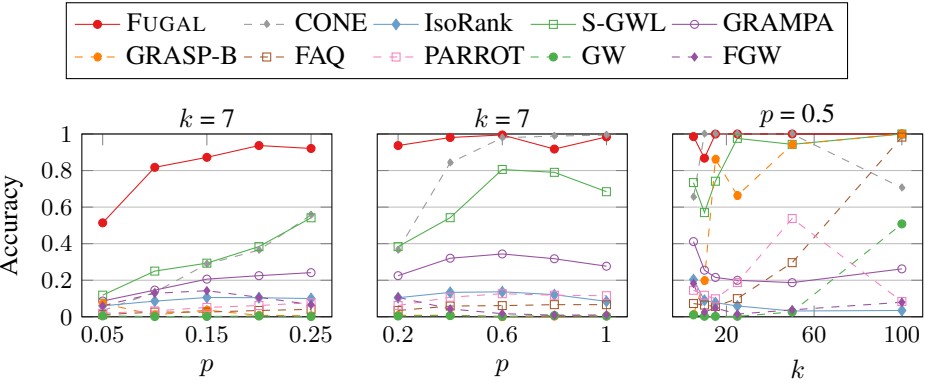

Figure 4: Accuracy varying density and one-way noise.

## 5.5 Efficiency

Here we compare the computational efficiency of FUGAL to that of S-GWL, which ranks as the second-best performer across most benchmark datasets.

Figure 5 plots running times *in logarithmic axes*. FUGAL achieves lower running times on MultiMagna, Voles, euroroad, arenas, and Newmann-Watts networks with an up to 3x speed up, highlighting its capacity to handle *large* networks. Conversely, S-GWL marginally outperforms FUGAL on *smaller* networks. S-GWL did not scale for ACM-DBLP, failing to terminate even after 5 hours. This discrepancy indicates S-GWL's incapacity to scale to large networks, which restricts its broader applicability. We emphasize

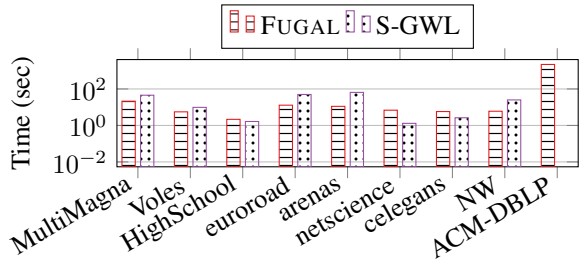

Figure 5: Running time comparison, FUGAL vs S-GWL.

that FUGAL achieves a substantial accuracy advantage without compromising efficiency, affirming its prowess as an efficient and effective solution. Appendix A.2 presents running times for all baselines.

## 5.6 Scalability

Given the results of Section 5.5, we delve into the scalability of FUGAL and S-GWL with Newmann-Watts graphs of increasing nodes. Figure 6 plots our findings. At 512 nodes, FUGAL and S-GWL have comparable running times. Still, as nodes grow, FUGAL outpaces S-GWL.

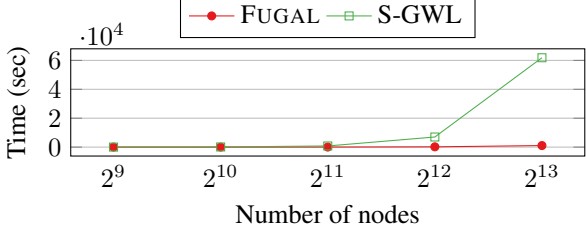

Figure 6: Scalability on NW graphs, $k = 7$, $p = 0.5$.

## 5.7 Parameters and Ablation

Table 3 in Appendix lists the parameters we employ in FUGAL with each dataset. We set the number of iterations $T$ to 15 for all datasets. The parameter $\mu$ controls the sway of node features in the optimization. Sparser graphs, characterized by lower connectivity and less information in adjacency matrices, benefit from higher reliance on node features, hence we recommend a higher $\mu$. Sparser datasets such as inf-euroroad, ca-netscience, and NW ($k = 7$, $p = 0.1$) benefit from higher values of $\mu$ (1–2), denser graphs from smaller values (0.1–0.5).

We also conduct an extensive ablation study to assess the impact of the structural features outlined in Section 3.1. We craft five variants of FUGAL, where FUGAL-$i$ utilizes only the $i^{\text{th}}$ structural feature while excluding others. FUGAL-0 abstains from all structural features. Figure 7 juxtaposes the accuracy of these variants to that of FUGAL on two networks. Each variant employing structural features attains higher accuracy than FUGAL-0, corroborating the usefulness of these features. Further, FU-GAL, leveraging all features, outperforms other variants. Notably, FUGAL-1 performs

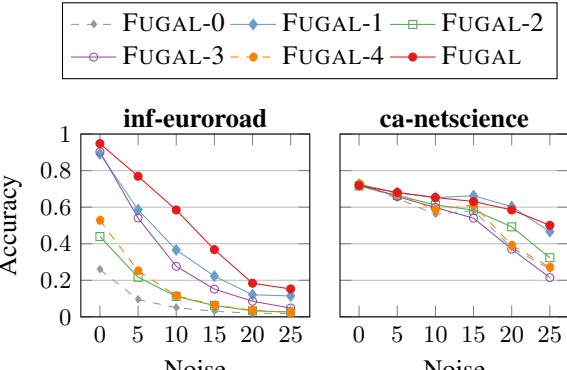

Figure 7: Accuracy of FUGAL variants, one-way noise.

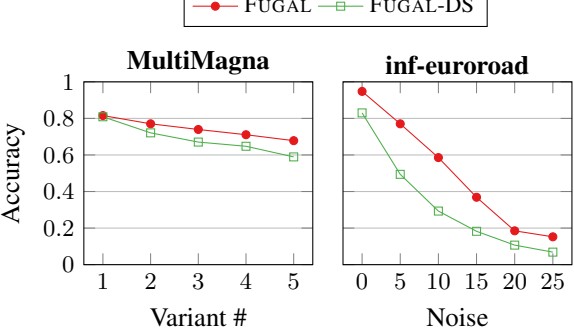

Figure 8: The effect of setting $\lambda = 0$ (FUGAL-DS) against the default option of iteratively increasing.

second-best, underscoring the significance of degree in identifying node alignments. We also investigate a variant setting $\lambda = 0$, denoted as FUGAL-DS (for *doubly stochastic*), instead of iteratively increasing it. As Figure 8 shows, FUGAL-DS attains worse accuracy.

## 6 Conclusions

We introduced FUGAL, an *unrestricted* algebraic approach to graph alignment that works directly on graph adjacency matrices and identifies node correspondences by relaxing permutation matrix constraints and steering the solution to the desired form, followed by rounding. Through extensive experimentation, we established that FUGAL surpasses state-of-the-art graph alignment methods in accuracy across network types, noise conditions, and graph densities, even while maintaining a scalability advantage.

**Broader Impact and Ethical consequences:** FUGAL opens the way to improved solutions in graph alignment, as reflected in its performance across diverse networks, noise types, and graph density. This outcome can spark further research in optimization techniques and advances in bioinformatics, social network analysis, and infrastructure mapping. Still, advances in graph alignment also enhance the abilities of attackers attempting to de-anonymize sensitive social network and biological data. Therefore, preventing attacks on privacy is crucial, calling for the enforcement of advanced anonymization methods [34] before publishing such data.

## 7 Acknowledgements

AB was supported by Graviton Research Capital LLP. HRV was supported by the CSE Research Acceleration Fund of IIT Delhi. KS was supported by the Independent Research Fund Denmark.

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

# A Appendix

---

**Algorithm 2** FUGAL $(\mathcal{G}_1, \mathcal{G}_2)$

---

**Input:** Graphs $\mathcal{G}_1, \mathcal{G}_2$
**Output:** Permutation matrix $\mathbf{P}$

1: \\ STEP 1. Extract NetSimile Features
2: $\mathbf{F}_1 \leftarrow$ EXTRACTFEATURES$(\mathcal{G}_1)$
3: $\mathbf{F}_2 \leftarrow$ EXTRACTFEATURES$(\mathcal{G}_2)$
4: $\mathbf{D} \leftarrow$ EUCLEDIANDISTANCE$(\mathbf{F}_1, \mathbf{F}_2)$
5: \\ STEP 2. Approximate Optimization
6: $\mathbf{Q} \leftarrow$ FINDQUASIPERMUTATION$(\mathbf{A}, \mathbf{B}, \mathbf{D}, \mu, T)$
7: \\ STEP 3. Round to Permutation
8: $\mathbf{P} \leftarrow$ HUNGARIAN$(\mathbf{Q})$
9: **return P**

---

Table 3: Parameters used in FUGAL.

| **Dataset** | $\mu$ |
|---|---|
| Arenas | 0.5 |
| inf-euroroad | 2 |
| bio-celegans | 0.1 |
| ca-netscience | 1 |
| MultiMagna | 0.5 |
| HighSchool | 0.5 |
| Voles | 0.5 |
| Newmann-Watts | 2 |
| ACM-DBLP | 0.1 |

## A.1 Complexity Analysis

To assess the computational complexity of FUGAL, we examine the three primary components: (i) structural feature extraction; (ii) obtaining a quasi-permutation matrix; (iii) rounding to a permutation matrix. Let us consider source and target graphs $\mathcal{G}_1, \mathcal{G}_2$ with $n$ nodes. Among NETSIMILE structural features, the clustering coefficient incurs a high computational cost of $\mathcal{O}(nM^2)$, where $M$ is the maximum degree among vertices in the graph. Still, for real-world graphs conforming to a power-law degree distribution, the complexity for neighborhood features extraction is expected to be $\mathcal{O}(nM^\epsilon)$, with $0 < \epsilon < 1$ [18], while finding pairwise Euclidean distances between node features takes $\mathcal{O}(n^2)$. The quasi-permutation matrix derivation involves determining the gradient of the optimization problem, which requires $\mathcal{O}(n^3)$ due to matrix multiplications. The Sinkhorn-Knopp algorithm finds the doubly-stochastic matrix $q$ minimizing $\langle grad, q \rangle$, is nearly $\mathcal{O}(n^2)$ [7], while the update to $Q$ takes $\mathcal{O}(n^2)$ time. Thus, the time complexity for finding a quasi-permutation matrix is $\mathcal{O}(T \cdot n^3)$, where $T$ is the number of iterations. We round the quasi-permutation matrix to a permutation matrix by the Hungarian algorithm, incurring a time complexity of $\mathcal{O}(n^3)$. Therefore, the time complexity of FUGAL is $\mathcal{O}(T \cdot n^3)$. Empirically, $T$ typically ranges from 10 to 20, resulting in a cost of $\mathcal{O}(n^3)$ since $T \ll n$. We provide computational costs of baselines in Appendix A.6. Table 4 compares FUGAL's computational cost to that of baselines.

## A.2 Running times for all baselines

Figure 9 presents the running times of various baselines on the benchmark datasets. It is noteworthy that while some of these baselines exhibit better running times than FUGAL, the substantial disparity in accuracy, as previously demonstrated in §. 5, renders a comparison skewed in favor of FUGAL.

## A.3 Accuracy on Real Graphs with Synthetic Noise - Low Noise Range

In § 5, we have already demonstrated the superior performance of FUGAL in the presence of noise levels ranging from 0% to 25%. In this section, we extend the comparison to assess the performance

Table 4: Computational Complexity comparison with baselines. $n, m$ denotes the number of nodes and edges respectively. $T, L$ denote the number of loop iterations.

| | CONE | IsoRank | S-GWL | GRAMPA | GRASP-B | FAQ | PARROT | fGOT | GOT | FUGAL | GW | FGW |
|---|---|---|---|---|---|---|---|---|---|---|---|---|
| **Time** $\mathcal{O}(.)$ | $n^2$ | $n^4$ | $(n+m)\log n$ | $n^3$ | $n^3$ | $n^3$ | $Tmn + TLn^2$ | $n^3$ | $n^3$ | $n^3$ | $n^3$ | $n^3$ |

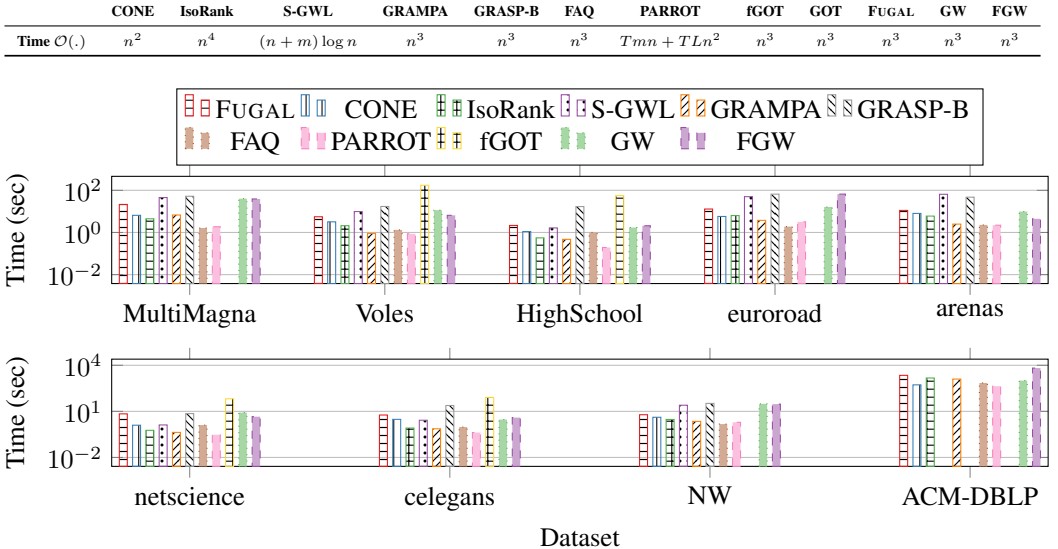

Figure 9: Running times of all baselines.

of FUGAL and other baseline methods on real datasets with synthetic noise, specifically of One-Way and Multi-Modal types, within the range of 0% to 5%. The results are presented in Figure 10. On the **inf-euroroad** dataset, FUGAL consistently outperforms all baselines across varying noise levels in both One-Way and Multi-Modal scenarios. For the **arenas** dataset, FUGAL, CONE, S-GWL, PARROT, and GW maintain accuracy levels exceeding 95% consistently across all noise levels, whereas other baselines exhibit a decline in performance with increasing noise levels, particularly evident in the Multi-Modal scenario. In the case of the **ca-netscience** dataset, both FUGAL and S-GWL achieve similar accuracy in both One-Way and Multi-Modal scenarios. On the **bio-celegans** dataset, S-GWL closely matches the performance of FUGAL in the One-Way scenario; however, with Multi-Modal noise, FUGAL achieves a 6% improvement over the next-best method, S-GWL. The consistent and superior performance of FUGAL across all benchmark datasets underscores its robustness.

## A.4 Evaluation on Small Graphs

To compare the performance of non-scalable methods like GOT, fGOT, PATH and DSPP with FUGAL, we employed small Erdős-Rényi random graphs [12]. The limited scalability of these methods for larger graphs is empirically demonstrated in Section 5 (failing to terminate within 5 hours) as well as evidenced by the maximum graph size evaluated by the authors, which was 100. Following the methodology of fGOT [32], we varied the node count $n$ from 20 to 100, with edges generated using a probability of $2\log(n)/n$. We also included S-GWL and PARROT in the analysis. The accuracy of these methods across different graph sizes is depicted in Figure 11. While FUGAL, S-GWL, and PARROT achieved perfect alignment across all graph sizes, other methods exhibited notably inferior performance. Following [32], the Frobenius distance between aligned graph Laplacian matrices across varying graph sizes is also reported in Figure 11. FUGAL, S-GWL, and PARROT maintained an L2 Distance of 0 across all graph sizes, indicating perfect alignment, whereas the performance of other methods deteriorated with increasing graph size. These distance values closely align with those reported by the original authors in [32], validating our experimental setup.

## A.5 Additional Metrics

We assess FUGAL against S-GWL and CONE in terms of Matched Neighborhood Consistency (MNC) [4] and the Frobenius distance between aligned graph adjacency matrices. The results in Figure 12 indicate that FUGAL outperforms S-GWL and CONE across noise levels and noisy variants.

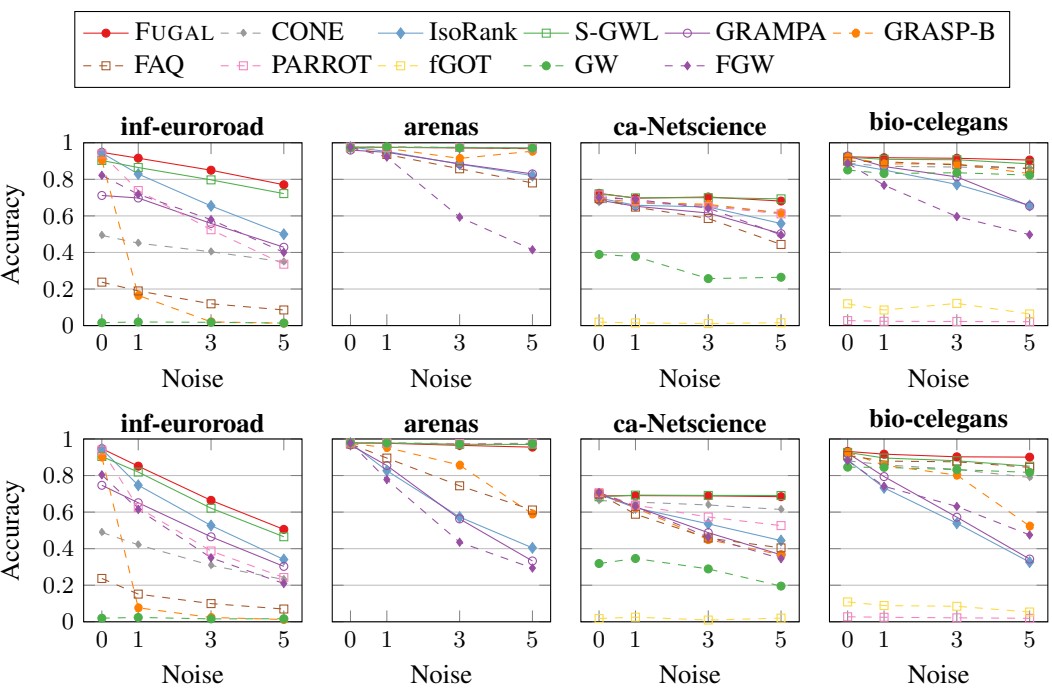

Figure 10: Accuracy comparison for real datasets with noise ranging from 0% to 5%. Top row represents One-Way noise, bottom row represents Multi-Modal noise scenarios.

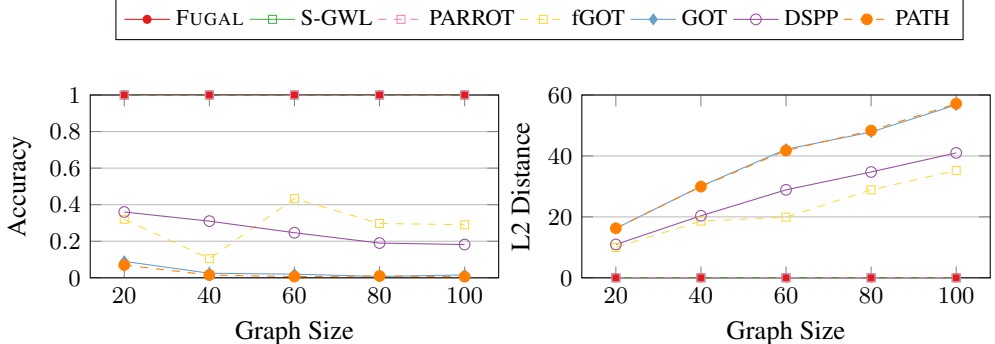

Figure 11: Performance comparison on Erdős-Rényi graphs. The performance is shown in terms of Accuracy (L) and the Frobenius distance between aligned graph Laplacian matrices (R) across different graph sizes (# of nodes).

## A.6 Computational Complexity Analysis of Baselines

The computational costs incurred by the baseline methods and FUGAL are presented in Table 4. It is notable that all algorithms utilize the Hungarian algorithm to convert the similarity matrix into a permutation matrix, incurring a computational cost of $\mathcal{O}(n^3)$. However, the costs reported in Table 4 solely pertain to the computation of the similarity matrix, excluding this operation. Among the baselines, CONE, S-GWL, and PARROT exhibit superior time complexity compared to FUGAL. However, as demonstrated in Section 5, both CONE and PARROT fall significantly short of FUGAL in terms of performance. Despite the seemingly promising computational cost of S-GWL, our empirical analysis in Sections 5.5 and 5.6 revealed slower running times compared to expectations. Moreover, S-GWL fails to scale for larger graphs such as ACM-DBLP (failing to terminate within 5 hours), whereas FUGAL achieves superior accuracy within 40 minutes. The limited scalability of S-GWL has been underscored by various studies [46, 19, 39, 28]. Consequently, FUGAL emerges as a preferable option for attaining superior accuracy without incurring detrimental overhead.

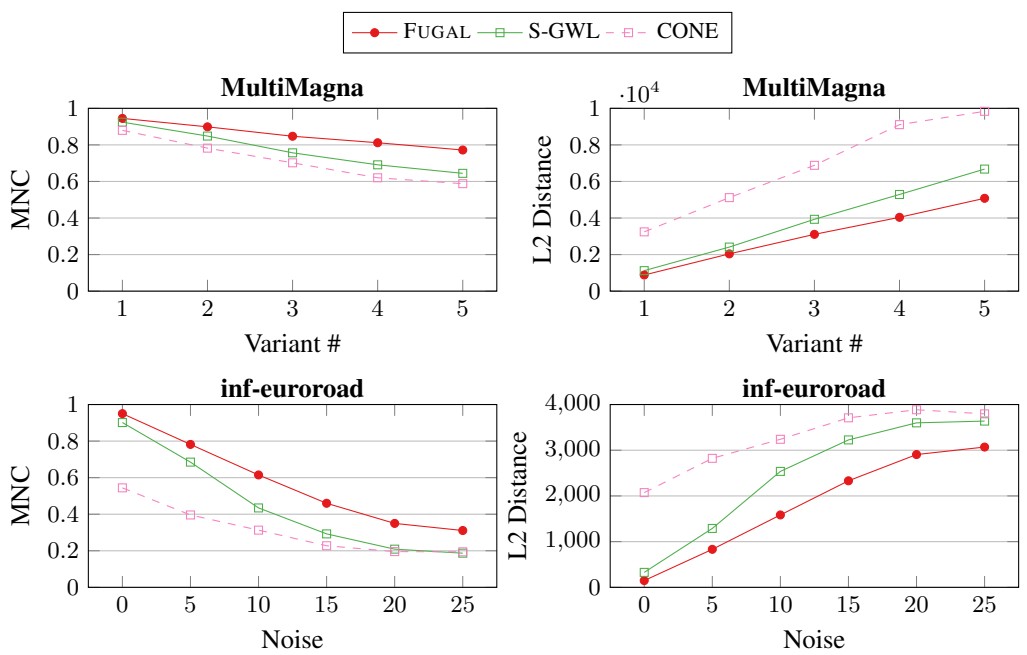

Figure 12: Comparison of FUGAL with S-GWL and CONE on MultiMagna and inf-euroroad datasets in terms of *Matched Neighborhood Consistency (MNC)* (L) and the *Frobenius distance* between aligned graph adjacency matrices (R) across different variants and varying noise respectively.

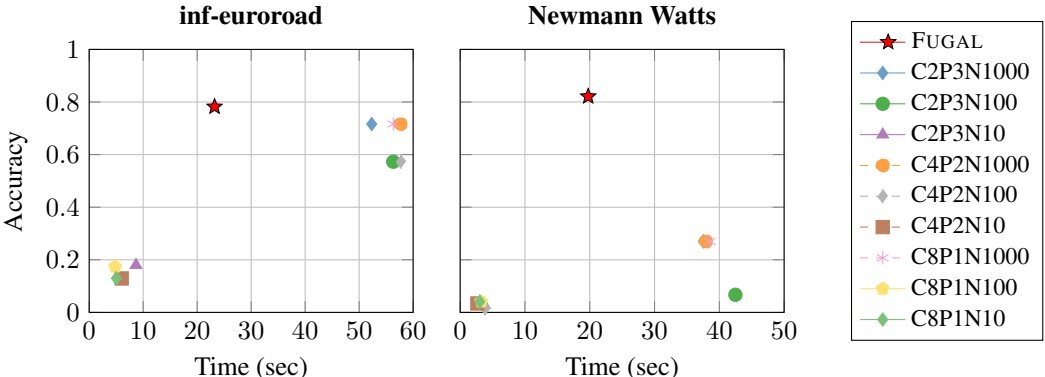

Figure 13: Accuracy vs. Running time, FUGAL vs S-GWL variants, one-way noise 5%.

## A.7 Accuracy vs Running Time

S-GWL employs a recursive graph partition mechanism to accelerate graph alignment computations. An important question is how the performance of SGWL is affected by this recursive mechanism with respect to FUGAL. We have identified that the hyper-parameters cluster_num (**C**), partition_level (**P**), and node_prior (**N**) significantly influence the recursive partitioning process. Three variants of S-GWL were proposed by the authors [43] based on different combinations of **C** and **P**: (**C** = 2, **P** = 3), (**C** = 4, **P** = 2), and (**C** = 8, **P** = 1). Additionally, we vary the node_prior parameter within the set {10, 100, 1000}. These combinations result in nine distinct variants of S-GWL, denoted as CxPyNz where x, y, z represent the values of **C**, **P**, and **N** respectively. We conduct a comparative evaluation of FUGAL against these nine variants with respect to both accuracy and running times jointly. Figure 13 depict the outcomes on the inf-euroroad and Newmann Watts datasets. For the inf-euroroad dataset, variants achieving comparable accuracy to FUGAL exhibit significantly higher running times, while those with lower running times have accuracy less than 20%. In the Newmann Watts dataset, none of the S-GWL variants approach the accuracy of FUGAL, as previously established in Section 5. Notably, variants with reasonable accuracy tend to have longer running times, whereas those with better running times demonstrate poorer accuracy. These findings underscore the practical suitability of FUGAL for graph alignment, given its favorable balance between accuracy and running times.

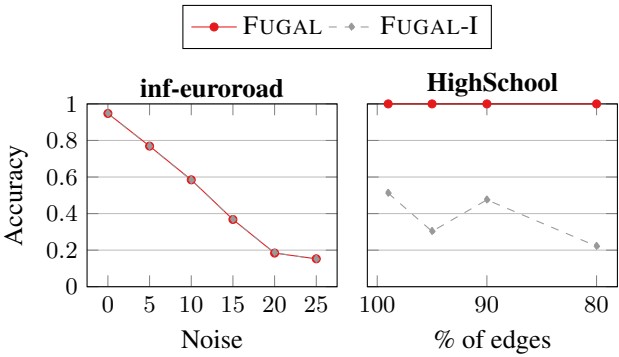

Figure 14: Accuracy, FUGAL vs FUGAL-I, one-way noise.

## A.8 Effect of Initialization

Lastly, we empirically examine our decision to employ a *non-informative* flat doubly stochastic matrix, denoted as $\mathbf{1} \cdot \mathbf{1}^T/n$, as the initial quasi-permutation matrix in Algorithm 1, in contrast to alternatives. We try out a variant of FUGAL, FUGAL-I, which utilizes the *Identity* matrix as the initial quasi-permutation matrix instead. Figure 14 presents two instances of our comparative evaluation of FUGAL-I vs FUGAL. While FUGAL-I closely reaches the accuracy of FUGAL on the **inf-euroroad** network, it falls short of FUGAL on the **HighSchool** network, indicating its instability. These findings substantiate our selection of the flat doubly stochastic matrix as a robust choice for initialization.

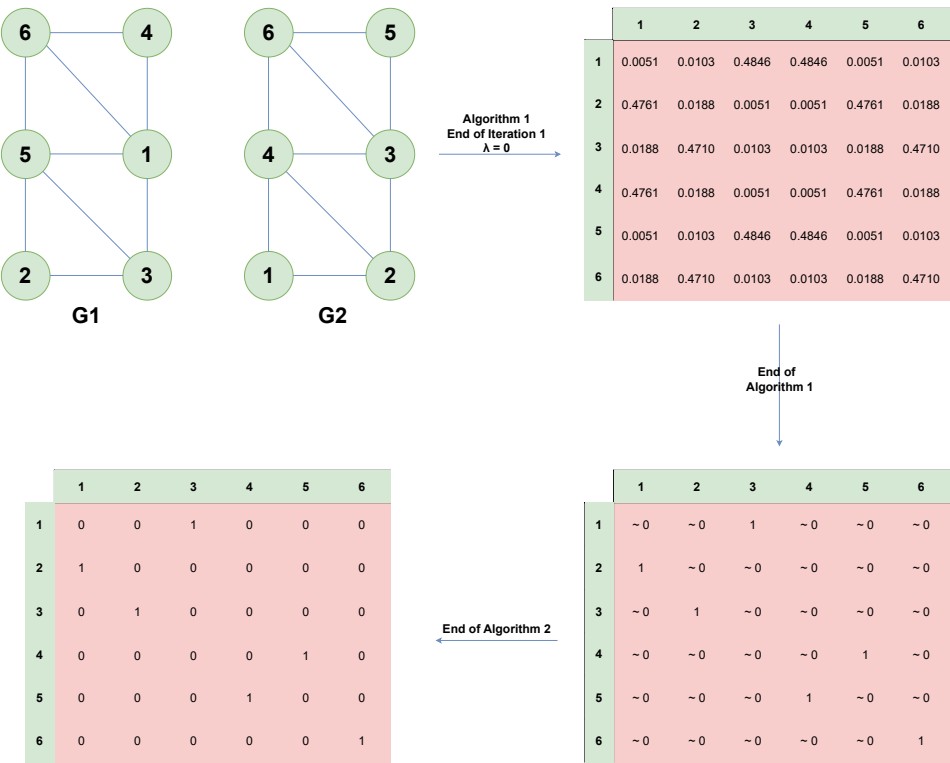

Figure 15: Operational stages of FUGAL. Values less than $1e^{-10}$ are approximated to 0.

## A.9 Illustrative example of FUGAL's pipeline

We illustrate the functionality of FUGAL through an example in Figure 15. We create a source graph $\mathcal{G}1$ of 6 nodes and randomly permute it to a target graph $\mathcal{G}2$. We show three stages of FUGAL's operation: (1) the doubly stochastic matrix $\mathbf{Q}$, generated after the first iteration of Algorithm 1 ($\lambda = 0$); (2) the quasi-permutation matrix $\mathbf{Q}$ to which Algorithm 1 (Section 3) steers the doubly stochastic matrix; (3) the permutation matrix $\mathbf{P}$ into which Algorithm 2 refines this quasi-permutation matrix using the Hungarian algorithm. FUGAL aligns $\mathcal{G}1$ and $\mathcal{G}2$ perfectly.

