# OpenReview forum: "FUGAL: Feature-fortified Unrestricted Graph Alignment"
_NeurIPS.cc/2024/Conference — NeurIPS 2024 poster_

### Official Review · Reviewer_ts95 · 2024-07-10

**Soundness:** 2
**Presentation:** 1
**Contribution:** 2
**Rating:** 5
**Confidence:** 5

**Summary:**

In this paper, the authors propose an algorithm for graph matching. The method is based on the classical relaxation to the set of doubly stochastic matrices, where the authors add two variants to the optimization: (1) an additional term to match node features (computed from the graph as degree, clustering coefficient, and others), and (2) an extra term to push the matrix to be closer to a permutation matrix. The paper presents some promising experimental evaluations.

**Strengths:**

The paper is well written in general. The strength of the paper is the combination of a classical formulation (the relaxation to doubly stochastic matrices, and I would add the features-term as classical as well) with a new term in the optimization, pushing the solution closer to a permutation matrix. Although the paper has no theorethical guarantees at all, it provides some experimental evaluation.

**Weaknesses:**

One huge weakness is the absolute lack of references to related work.
In the paper (line 57), the authors say that FAQ is the only algorithm addressing the graph matching (they say the QAP, actually) problem directly through the adjacency matrices. This is profoundly inaccurate.
Here are some references, some of them with formulations extremely related to the one presented in this paper. Most of them pose the problem by relaxing to the set of doubly stochastic matrices. Some of them add a term for "feature matching" (like [1] and [3], equation (21) in [1] adds exactly the same term as in eq (7) in this paper). And some of them add an extra term pushing the doubly stochastic matrix closer to a permutation matrix (as presented in this paper), for instance [1] and [6].

[1] Zaslavskiy, M., Bach, F., & Vert, J. P. (2008). A path following algorithm for the graph matching problem. IEEE Transactions on Pattern Analysis and Machine Intelligence, 31(12), 2227-2242.

[2] Fiori, M., Sprechmann, P., Vogelstein, J., Musé, P., & Sapiro, G. (2013). Robust multimodal graph matching: Sparse coding meets graph matching. Advances in neural information processing systems, 26.

[3] Zhou, F., & De la Torre, F. (2015). Factorized graph matching. IEEE transactions on pattern analysis and machine intelligence, 38(9), 1774-1789.

[4] Aflalo, Y., Bronstein, A., & Kimmel, R. (2015). On convex relaxation of graph isomorphism. Proceedings of the National Academy of Sciences, 112(10), 2942-2947.

[5] Fiori, M., & Sapiro, G. (2015). On spectral properties for graph matching and graph isomorphism problems. Information and Inference: A Journal of the IMA, 4(1), 63-76.

[6] Nadav Dym, Haggai Maron, Yaron Lipman. (2017) Ds++: A flexible, scalable and provably tight relaxation for matching problems


This makes the experimental section kind of weak also, since at least I would have compared with [1] and eventually [6].

Besides that, to me the experimental section lacks an experiment showing the gain from each addition to the vanilla formulation. This is, what is the difference in performance with and without the features, and lambda=0 vs other lambda.

Another weakness is the lack of theoretical results in terms of guarantees of the method, or at least a comment on which part has or has not guarantees. For instance, the formulation is guaranteed to obtain the solution for some graphs? (in the spirit of Aflalo et. al [4]). The optimization algorithm is guaranteed to obtain a local minima?
I understand the properties of the FW method, but I'm not sure that the presented algorithm solves exactly every sub-process of the FW method.

Another comment is the way of measuring the error. The authors measure the coincidence between the obtained permutation matrix and "the" true isomorphism. However, there may be many solutions, if the graph has nontrivial automorphism group (hence the "the" in the previous sentence). This is the case for instance of the graph presented in the paper in Figure 13.
A more suitable measure could be $||AP-PB||_F^2$

**Questions:**

In Algorithm 1, the number of iterations T is the value for lambda??
Why does lambda vary in the integers? And how is the value of lambda chosen?

When you change the problem of finding the q minimizing the inner product with grad, with the formulation in eq (16), the solution is not guaranteed to be the same.
How do you choose the lambda in this case? (also, not a good idea to use the same name for different parameters)

In line 179, the rounding process is the same as projecting the solution to the set of permutation matrices? Because this projection is solved exactly as an LAP with the Hungarian algorithm for example.
Also, why use the Hungarian algorithm and not the Sinkhorn formulation here? It is the same problem.

Minor comment:
In line 33, there's a word missing (method?) "In this work, we propose an unrestricted graph alignment m that avoids restricting ..."

**Limitations:**

There's no explicit mention of any limitation of the algorithm.

---

> ### Author Rebuttal · Authors · 2024-08-05
>
> **W1(a). Lack of references: the authors say that FAQ is the only algorithm addressing the graph matching (they say the QAP, actually) problem directly through the adjacency matrices. This is profoundly inaccurate. [1]-[6]**
>
> **Answer:** We apologize for the lack of precision in our original statement. Our intent was to convey that FAQ is one of the few algorithms that solves QAP for network alignment **without** introducing additional regularizers, which we term the *unmediated* approach. As defined in footnote 1, when QAP is augmented with mediated features, such as in our regularizer and several other algorithms (e.g., PATH [1] using feature matching, [6] employing all pairs shortest paths, [3] with feature matching, and [4] utilizing vector attributes), they fall into *unrestricted* category.
>
> We acknowledge that our initial claim about FAQ being the only unmediated approach was incorrect. GLAG [2] is also an unmediated approach and we will revise our statement accordingly. Besides, [7] shows that GLAG's relaxation leads to worse alignments than FAQ.
>
> [7] Lyzinski, V. et al. Graph matching: Relax at your own risk. PAMI, 2015.
>
> **W1(b) Compare with [1] and [6].**
>
> **Answer:** We have incorporated PATH[1] and DDSP[6]. `Fig. 1 in the pdf attached to the global rebuttal` presents the results. Both baselines fall short compared to FUGAL. Furthermore, as the graph sizes grow, the performance gap widens.
>
> Additionally, both algorithms are prohibitively slow and fail to return an alignment within 5 hours even on the smallest real-world dataset in our experiments, ca-netscience. Hence, we report results on synthetic datasets. [7] (cited above) have shown FAQ to outperform Path.
>
> **W2. What is the difference in performance with and without the features, and lambda=0 vs other lambda.**
>
> **Answer:** We already include experiments on feature ablation study and the impact of setting both $\lambda=0$ and $\mu=0$. We have now also added the third ablation for $\lambda=0$.
>
> * **Features:** We have a detailed ablation study in Appendix A.3 (referred from Section 5.2 in main manuscript), where we systematically turn on each feature and study its impact (Fig 8). The results reveal that degree is the most important feature, followed by the mean degree of neighbors.
>
> * **$\lambda=0$**: `Fig. 3 in the pdf attached to the global rebuttal` presents the results. If $\lambda$ is not iteratively increased (recall Alg. 1), the performance suffers.
>
> **W3(a). Is the formulation guaranteed to obtain the solution for some graphs? (like [4]).**
>
> **Answer:** We do not have guarantees of an optimal solution for any specific class of graphs.
>
> **W3(b). Is the optimization algorithm guaranteed to obtain a local minima?**
>
> **Answer:** The problem is not convex due to the regularizer that guides the solution towards a permutation matrix. In this scenario, we do not guarantee obtaining local minima. However, as discussed in W2, the proposed strategy of iteratively increasing $\lambda$ yields better accuracy.
>
> **W3(c). I'm not sure that the presented algorithm solves exactly every sub-process of FW.**
>
> **Answer:** It does. Each iteration of the algorithm uses FW within which we employ optimal transport to determine the doubly-stochastic matrix $q$ that minimizes the inner product with $grad$. However, across iterations, we increment $\lambda$ to guide the solution towards a permutation matrix.
>
> **W4. ...A more suitable measure could be $||AP-PB||_F^2$.**
>
> **Answer:** We have added *Frobenius Norm*, as well as, *matched neighborhood consistency (MNC)* as additional metrics. The results are presented in `Fig. 2 of the PDF attached to the global response`. Consistent with the accuracy results, FUGAL maintains its competitive edge.
>
> **Q1. In Alg. 1, the number of iterations T is the value for lambda?? Why does lambda vary in the integers? And how is the value of lambda chosen?**
>
> **Answer:** Yes, $\lambda$ varies from 0 to $T-1$ in integers. We set $T$ to $15$ across all datasets. Varying $\lambda$ in the integers is just an empirical decision; it's not a constraint. Both $T$ and the granularity of increasing $\lambda$ are hyper-parameters. We choose $T=15$ since at this value FUGAL demonstrates robust accuracy across all datasets.
>
> **Q2. When you change the problem of finding the $q$ minimizing the inner product with grad, with the formulation in eq (16), the solution is not guaranteed to be the same. How do you choose the lambda? (also, not a good idea to use the same name for different parameters)**
>
> **Answer:** We set $\lambda=1$ across all datasets. We will update the variable name and mention this hyperparameter setting in  App. A.3.
>
> **Q3(a). In line 179, the rounding process is the same as projecting the solution to the set of permutation matrices? Because this projection is solved exactly as an LAP with the Hungarian algorithm.**
>
> **Answer:** Our rounding algorithm employs maximum weight matching using the Hungarian method. We construct a complete bipartite graph, with nodes from graphs A and B forming the two partite sets. Edge weights correspond to the values in the quasi-permutation matrix produced by Alg. 1. The Hungarian algorithm determines the optimal one-to-one mapping between the node sets, maximizing the cumulative weight of the selected edges.
>
> **Q3(b). why use the Hungarian and not Sinkhorn here?**
>
> **Answer:** Sinkhorn projects the matrix into the space of doubly stochastic matrices. But we need a permutation matrix as output, hence the Hungarian.
>
> **Q4. In line 33, there's a word missing (method?)**
>
> **Answer.** We'll correct this.
>
> ----------
>
> # Appeal to the reviewer
>
> Thank you for helping us with actionable comments on our work. We have comprehensively incorporated _all_ suggestions by adding new baselines, ablation studies, metrics, and clarifications. We would be grateful if the reviewer could reassess our paper in light of these improvements and consider adjusting the rating accordingly.

---

> > ### Author Response · Authors · 2024-08-13
> > **Eagerly awaiting feedback from Reviewer ts95**
> >
> > Dear Reveiwer ts95,
> >
> > First of all, thank you for taking your time out in reviewing our work and providing constructive feedback. Based on your suggestions, we have incorporated new baselines, ablation studies, metrics. We are eagerly awaiting your feedback on the rebuttal. We humbly appeal to you to please review the revisions made since the discussion phase closes in less than 2 days from now.
> >
> > regards,
> >
> > Authors.

---

> > > ### Author Response · Authors · 2024-08-13
> > > **Keenly awaiting feedback from Reviewer ts95**
> > >
> > > Dear Reviewer ts95,
> > >
> > > We are less than a day away from the closure of the author-reviewer discussion phase. We are keenly awaiting your feedback on our detailed rebuttal. We thank you for your constructive feedback and hope all your concerns have now been addressed.
> > >
> > > regards,
> > >
> > > Authors

---

> > ### Comment · Reviewer_ts95 · 2024-08-13
> >
> > I want to thank the authors for the detailed response.
> >
> > Some of my concerns were taken, as well as suggestions. I'll raise my score accordingly.
> >
> > I still think that the heuristic optimization needs a more precise framework, and a stronger theoretical study.

---

### Official Review · Reviewer_FAZB · 2024-07-11

**Soundness:** 3
**Presentation:** 3
**Contribution:** 3
**Rating:** 7
**Confidence:** 4

**Summary:**

The paper proposes FUGAL, a method for graph alignment by using additional features of nodes to guide optimization of a relaxed problem. It combines strengths of 2 lines of methods, using full graph information and structural features enrichment.

---
score raised after rebuttal.

**Strengths:**

I think the paper blends the approach in a reasonable way to take the advantages of mediated  and unmediated approaches. The idea seems right and its formulation seems sound. It has very good performance to back up the choices in the method.

**Weaknesses:**

This is not a totally novel approaches but it is understandable given the nature of the combinatorial optimization problem.

There should be a lot of room for improvement in terms of the structural features used in the paper. There might be many more to choose from. Depending on the nature of the application domain, some feature sets may make sense more than other. Some additional analysis on this part might be interesting.

**Questions:**

N/A

---

> ### Author Rebuttal · Authors · 2024-08-05
>
> We thank the reviewer for the positive comments on our work. Please find below some clarifications on the queries posed.
>
> **W1. Clarification on novelty**
>
> *Answer.* The novelty of our work lies in:
> 1. crafting a regularizer using network features that makes the QAP potent for network alignment. State of the art algorithms for network alignment had generally moved away from a QAP based formulation due to its non-competitive performance
> 2. Devising a novel optimization strategy wherein we guide the Frank-Wolfe algorithm through a Sinkhorn distance objective, and gradually steer the resulting doubly stochastic solution towards a quasi-permutation matrix.
>
> With these innovations, we design an algorithm that outperforms 13 different baselines in a comprehensive empirical benchmarking exercise and achieves state-of the-art-performance.
>
> **W2. There should be a lot of room for improvement in terms of the structural features used in the paper. There might be many more to choose from. Depending on the nature of the application domain, some feature sets may make sense more than other. Some additional analysis on this part might be interesting.**
>
> *Answer.* We'd like to draw your attention to Appendix A.3 (referenced in Section 5.2 of the main manuscript), where we've already included a comprehensive ablation study examining the importance of each feature (Fig. 8). Our findings indicate that degree is the most crucial feature, closely followed by the mean degree of neighbors.
>
> We concur with your insight that identifying other beneficial structural features remains an open research question. This point is explicitly addressed in lines 128-131 of our manuscript. To further supplement this analysis with empirical data, we further expanded our feature ablation study by covering more node features spanning PageRank (PR), Eigen Centrality (EC) and Closeness Centrality (CC).
>
> The table below compares the efficacy of using only one of these features as opposed to the 4 vanilla features used in Fugal (degree, mean degree of neighborhood, clustering coefficient, mean clustering coefficient in neighborhood). The experiment is performed on the ca-netscience dataset at various noise levels. Among individual features, we note Degree to have the maximal impact and closely followed by PageRank. This result is not surprising given that PageRank is correlated to degree. Overall, Fugal, which uses 4 features, continues to be superior to relying on any single feature.
>
> #### **Caption**: Accuracy in the ca-netscience dataset at various noise levels. Fugal represents the default version that uses the four features mentioned above (and in Section 3.1). The other columns represent the accuracy achieved when only the corresponding feature is used in the LAP regularizer.
>
> Noise	| CS	| Degree|	EC|	PR	| Fugal
> ---|---|---|---|---|--
> 0|	0.70	|0.71|	0.70	|0.70	|0.70
> 5|	0.67	|0.70|	0.67	|0.69	|0.68
> 10|	0.57	|0.67|	0.61	|0.64	|0.67
> 15|	0.52	|0.63|	0.51	|0.55	|0.61
> 20|	0.34	|0.56|	0.39	|0.53	|0.57
> 25|	0.35|	0.48|	0.27|	0.46|	0.53
>
> The effectiveness of features can vary across datasets due to differences in network properties. Automatically selecting optimal features would require shifting to a supervised pipeline. We must also consider feature interactions, as individually informative features may not provide any marginal improvement when combined. The optimal feature set needs to account for potential signal overlap. However, this approach would demand training on each specific dataset, substantially increasing our framework's computational demands. This dataset-specific optimization, while potentially more accurate, would significantly complicate the overall process.
>
> We will include the above discussion in our revision.
>
> -------------------
>
> # Appeal to the reviewer
>
>  If the reviewer feels satisfied with the clarifications provided, we would appreciate support for our work by adjusting the rating accordingly.

---

> > ### Comment · Reviewer_FAZB · 2024-08-13
> >
> > I appreciate the rebuttal. I can see that the paper's contribution is pretty solid. I raise my score to reflect my current assessment of the paper.

---

> > > ### Author Response · Authors · 2024-08-13
> > >
> > > Thank you for your constructive feedback, which has helped us improve our work. We appreciate your support and the Accept rating.
> > >
> > > Sincerely,
> > >
> > > The Authors

---

### Official Review · Reviewer_b2PL · 2024-07-14

**Soundness:** 3
**Presentation:** 2
**Contribution:** 3
**Rating:** 7
**Confidence:** 3

**Summary:**

The paper presents FUGAL, a method for aligning graphs by finding a permutation matrix. FUGAL is an unrestricted method as it (also) operates on adjacency matrices, unlike most methods that rely only on intermediary graph representations.
FUGAL combines a Quadratic Assignment Problem (QAP) with a Linear Assignment Problem (LAP). The QAP focuses on finding the permutation matrix directly on the adjacency matrices, while the LAP works on node feature vectors built using structural features. Essentially, FUGAL augments the QAP with a LAP regularizing term. An initial algorithm is defined to find a quasi-permutation matrix  Q . Then, the authors propose refining  Q  using the Hungarian algorithm. Interestingly, FUGAL relaxes the solution space to doubly stochastic matrices.

**Strengths:**

The paper is well-written and the method is well-presented, especially in Section 3. The idea of combining QAP and LAP leads to an elegant solution, and the fact that it does not use intermediary graph representations distinguishes it significantly from other methods.

**Weaknesses:**

The authors do not clearly define the limitations of their method, and there is no clear sensitivity analysis or ablation study of certain aspects.
Besides, there is a complete lack of statistical significance analysis.

**Questions:**

Here are some suggestions and comments:

- Probably, a paragraph summarizing those cases in which the method works well and when it might perform poorly is missing. Maybe I missed something, but sections 5 and 6 should include the limitations as per your “NeurIPS Paper Checklist,” though they might not be well emphasized.

- In Chen, X., Heimann, M., Vahedian, F., & Koutra, D. (2020, October). “Cone-align: Consistent network alignment with proximity-preserving node embedding.” In Proceedings of the 29th ACM International Conference on Information & Knowledge Management (pp. 1985-1988) [CONE], there is an experiment on MNC and matched neighborhood consistency. Specifically, I refer to Figure 4. I suggest conducting a similar experiment, even if just to add in the appendix for further comparison.

- In Figure 1, using the variants of MultiMagna instead of $q$ may confuse the reader slightly.

- I wonder if the multimagna dataset you used corresponds to the PPI dataset used, for example, in [CONE] and also used in Xu, H., Luo, D., & Carin, L. (2019). “Scalable Gromov-Wasserstein learning for graph partitioning and matching.” Advances in neural information processing systems, 32.

If PPI is multimagna, in other works, such as in [CONE], I see that they used the average accuracy with standard deviation in error bars. You mentioned that “The error bars are compromising the visual interpretability of the plots due to the large number of baselines compared against,” but these values are significant, so at least in the appendix, I would suggest to visualize some by rearranging the plots.

- Regarding sensitivity analysis, I refer, for example, to the structural features. It is possible that only one of these features is actually useful or very influencing. I understand that this is not the focus of your method, but it would be interesting to see the influence of these node features for completeness.

**Limitations:**

Including some sentences that clearly stree the method's limitations would improve the paper.

---

> ### Author Rebuttal · Authors · 2024-08-05
>
> **Q1. Probably, a paragraph summarizing those cases in which the method works well and when it might perform poorly is missing. Maybe I missed something, but sections 5 and 6 should include the limitations as per your “NeurIPS Paper Checklist,” though they might not be well emphasized.**
>
> *Answer:* Our discussion on the limitations of FUGAL is interspersed in the discussion in Sections 5 and 6. These include:
> - In _line 325-326_, we point out that S-GWL is more efficient that FUGAL on small graphs.
> - In _line 566_, we complement that CONE, S-GWL, and PARROT exhibit superior time complexity compared to FUGAL.
> - In _Section 6_, we highlight the ethical aspect that advances in graph alignment also enhance the abilities of attackers attempting to de-anonymize sensitive network data.
>
> As suggested, we will collect them into a single paragraph to emphasize them more prominently.
>
> **Q2. In Chen, X., Heimann, M., Vahedian, F., & Koutra, D. (2020, October). “Cone-align: Consistent network alignment with proximity-preserving node embedding.” In Proceedings of the 29th ACM International Conference on Information & Knowledge Management (pp. 1985-1988) [CONE], there is an experiment on MNC and matched neighborhood consistency. Specifically, I refer to Figure 4. I suggest conducting a similar experiment, even if just to add in the appendix for further comparison.**
>
> *Answer:* As suggested, we have added *matched neighborhood consistent (MNC)* and *Frobenius Norm* as additional metrics. The results are presented in `Fig. 2 of the PDF attached to the global response`. These new metrics align with our previous findings using the accuracy metric, further vindicating FUGAL's superior performance across multiple evaluation criteria.
>
> **Q3. In Figure 1, using the variants of MultiMagna instead of $q$ may confuse the reader slightly.**
>
> *Answer:* We will change the $x$-axis to $q$ as suggested.
>
>  **Q4. I wonder if the multimagna dataset you used corresponds to the PPI dataset used, for example, in [CONE] and also used in Xu, H., Luo, D., & Carin, L. (2019). “Scalable Gromov-Wasserstein learning for graph partitioning and matching.” Advances in neural information processing systems, 32.**
>
>  *Answer:* Yes, these two refers to the same dataset.
>
> **Q5. If PPI is multimagna, in other works, such as in [CONE], I see that they used the average accuracy with standard deviation in error bars. You mentioned that “The error bars are compromising the visual interpretability of the plots due to the large number of baselines compared against,” but these values are significant, so at least in the appendix, I would suggest to visualize some by rearranging the plots.**
>
> *Answer:* Thanks for the suggestion. We will add standard deviation in the appendix.
>
> **Q6. Regarding sensitivity analysis, I refer, for example, to the structural features. It is possible that only one of these features is actually useful or very influencing. I understand that this is not the focus of your method, but it would be interesting to see the influence of these node features for completeness.**
>
> *Answer:*  We already include experiments on feature ablation study and the impact of setting both $\lambda=0$ and $\mu=0$ in Eq. 13. We have now also added the third ablation of only $\lambda=0$. Specifically:
>
> * **Features:** We have a detailed ablation study in Appendix A.3 (referred from Section 5.2 in main manuscript), where we systematically turn on each feature and study its impact (Fig 8). The results reveal that degree is the most important feature, followed by the mean degree of neighbors.
>
> * **$\lambda=0$ and $\mu=0$:** This setting degenerates to the FAQ method, already present in our experimental evaluation. As evident from our experiments, the deterioration in efficacy is significant.
>
> * **$\lambda=0$:** We have added this experiment now. `Fig. 3 in the pdf attached to the global rebuttal` presents the results. We see clear evidence that if $\lambda$ is not iteratively increased (recall Alg. 1), the performance is compromised.
>
> ------------
>
> # Appeal to the reviewer
>
> With the inclusion of additional metrics, ablation studies and clarifications, we hope the reviewer finds our manuscript improved. If the reviewer agrees, we would appreciate support for our work by increasing the rating accordingly.

---

> > ### Comment · Reviewer_b2PL · 2024-08-13
> >
> > I appreciate your detailed rebuttal. It has helped clarify several points.

---

### Official Review · Reviewer_T3ao · 2024-07-28

**Soundness:** 2
**Presentation:** 2
**Contribution:** 1
**Rating:** 3
**Confidence:** 4

**Summary:**

The current work tackles the problem of graph alignment, where the objective is to find an optimal alignment between two graphs. The current work attempts an unrestricted approach by solving a QAP and augmenting it with a LAP regularizer for tractability. This is in contrast to past work where the matching happens in an embedding space or an intermediate representation of the original graph, which incurs loss of information and consequently reduces the performance. The QAP finds a permutation matrix directly on the adjacency matrices of both the graphs and the augmented LAP uses structural features of the nodes to enhance node similarity of the matchings.

**Strengths:**

- The problem is aimed at solving an important problem of graph alignment
- The paper is clearly written
- I checked out the code, it is quite clean

**Weaknesses:**

There are several weaknesses of the paper:

1. I think the QAP problem has been well studied long before since 1990. See for example: https://www.math.cmu.edu/users/af1p/Texfiles/QAP.pdf.
https://link.springer.com/chapter/10.1007/978-1-4757-3155-2_6
https://link.springer.com/article/10.1007/s12652-018-0917-x
The authors need to perform comprehensive comparison against such approximation algorithm literature, since this paper does fall in the domain of improving the optimization of QAP. Otherwise, this paper does not add significant value.


2. One of the big advantage of embedding based graph alignment is during *test time* one does not have to re-optimize for alignment.
While yes, data driven methods will never be as strong as procedural algorithms, but on the flip side, the procedural algorithm have to re-run for unseen graphs as well.  I was expecting a comprehensive analysis to investigate the trade off. Just reporting time and accuracy won't help. One needs to see the curve between accuracy and time.

**Questions:**

See above.

**Limitations:**

I stated the key limitations in the weakenesses.

---

> ### Author Rebuttal · Authors · 2024-08-05
>
> **W1. I think the QAP problem has been well studied long before since 1990. See for example: https://www.math.cmu.edu/users/af1p/Texfiles/QAP.pdf.
> https://link.springer.com/chapter/10.1007/978-1-4757-3155-2_6 https://link.springer.com/article/10.1007/s12652-018-0917-x The authors need to perform comprehensive comparison against such approximation algorithm literature, since this paper does fall in the domain of improving the optimization of QAP. Otherwise, this paper does not add significant value.**
>
> *Answer:* The comments appears to step from a misunderstanding on the objective . **The objective of our work is not approximating QAP**. The objective **is network alignment**, which, among other formulations, can be formulated as an instance of QAP. Hence, we are not aiming to provide the best approximation for any given QAP, but for the specific network alignment instances. Our comprehensive empirical comparison already involves comparison with _11 state-of-the-art baselines_ for the network alignment problem, as evident from this survey [3]. On the other hand, due to the QAP formulation our study already included the FAQ baseline, that adopts the QAP approach for network alignment. Yet, FAQ fails to achieve competitive performance. To further shed light between graph alignment and QAP, we have added two more QAP-based baselines for network alignment [1, 2]. The experiments reveal the same pattern (See Fig. 1 in the pdf attached to global response), i.e., vanilla QAP formulations attain significantly worse results in graph alignment.
>
> The *value* of our work lies designing effective regularizers, that leverage and integrate various network-based features such as degree, clustering coefficient, etc., to the vanilla QAP, and a customized optimization strategy that empowers it to **achieve state of the art performance for network alignment.**
>
> We do not claim to have designed a state of the art QAP approximation algorithm. Rather, we assert that our method, which combines QAP with carefully selected network-based regularizers and optimization strategies, achieves state-of-the-art performance specifically for network alignment problems.
>
> [1] Zaslavskiy, M., Bach, F., & Vert, J. P. (2008). A path following algorithm for the graph matching problem. IEEE Transactions on Pattern Analysis and Machine Intelligence, 31(12), 2227-2242.
>
> [2] Nadav Dym, Haggai Maron, and Yaron Lipman. 2017. DS++: a flexible, scalable and provably tight relaxation for matching problems. ACM Trans. Graph. 36, 6, Article 184 (December 2017).
>
> [3] Konstantinos Skitsas, Karol Orlowski, Judith Hermanns, Davide Mottin, and Panagiotis Karras., Comprehensive evaluation of algorithms for unrestricted graph alignment. EDBT 2023.
>
> **W2. One of the big advantages of embedding based graph alignment is during test time one does not have to re-optimize for alignment. While yes, data driven methods will never be as strong as procedural algorithms, but on the flip side, the procedural algorithm have to re-run for unseen graphs as well. I was expecting a comprehensive analysis to investigate the trade off. Just reporting time and accuracy won't help. One needs to see the curve between accuracy and time.**
>
> *Answer:* To our knowledge, no embedding-based method has demonstrated competitive performance in network alignment. Moreover, supervised embedding approaches typically require dataset-specific training. This raises questions about the feasibility of performing inference on unseen test graphs without retraining. We would welcome information about any published, peer-reviewed algorithm that contradicts these observations. If such work exists, we would be happy to compare.
>
> **Accuracy-time trade-off:** The accuracy-time tradeoff against S-GWL, which is well established as the state-of-the-art, is already included in Fig. 11 along with a detailed discussion in Appendix A.7. At best, S-GWL achieves 20% lower accuracy when restricted to the running time of FUGAL. In addition, it never approaches the same accuracy as FUGAL.
>
> -------
>
> # Appeal to the reviewer
>
> We hope that our clarification on the scope and objectives of our work has provided a better context for our research. In response to the comments received, we have made several significant enhancements to our manuscript:
>
> 1. Added two more QAP-based baselines [1,2]
> 2. Introduced new metrics to provide a more robust assessment of performance
> 3. Included additional ablation studies to deepen the understanding of our method's components
>
> These empirical enhancements further substantiate our claim of advancing the state of the art in network alignment. In light of these improvements, we kindly request you to reassess our work and consider adjusting the rating accordingly. We would be glad to engage in further discussion during the author-reviewer period if you have any additional concerns.

---

> > ### Author Response · Authors · 2024-08-13
> > **Eagerly awaiting feedback on our rebuttal**
> >
> > Dear Reviewer T3ao,
> >
> > We are less than a day away from the closure of the discussion phase. We would greatly appreciate you feedback on the clarifications offered in our rebuttal. We are also happy to inform that two reviewers have already rated our work as "Accept (7)".
> >
> > regards,
> >
> > Authors

---

> ### Comment · Reviewer_T3ao · 2024-08-13
> **Response**
>
> Thanks to the authors for the rebuttal. My responses are as follows:
>
> > The objective of our work is not approximating QAP. The objective is network alignment, which, among other formulations, can be formulated as an instance of QAP. Hence, we are not aiming to provide the best approximation for any given QAP, but for the specific network alignment instances.
>
> A key application of QAP is network alignment.
> See for example:
> (1) https://arxiv.org/abs/1908.00265
> (2) https://journals.plos.org/plosone/article?id=10.1371/journal.pone.0121002
>
> I do not understand why a generic QAP solver will not work on the current problem.
> QAP is often given in the form of:
>
> $$ \max _{P} \text{trace}(MPQPR) $$ where $M,Q,R$ are known matrix and $P$ is a permutation matrix. Conceptually, the above problem can be easily cast into the above term (plus the linear term Tr(P^T D)).  Except the linear term, there is no difference from any general QAP problem. In fact, the current algorithm is not treating A, B etc., beyond ordinary matrices--- so there is nothing specific in the context of graphs in the underlying solution.
>
> In fact, the formulation is also almost same here: Eugene: explainable unsupervised approximation of graph edit distance https://arxiv.org/abs/2402.05885
>
> The novelty factor of this paper is too limited. In my view, the work is a QAP problem (with network alignment application), solved using a numerical optimization method (no learning involved) with some modification of very known algorithm (Sinkhorn/knopp, already used in GWL paper).
>
>
>
>
> >  To our knowledge, no embedding-based method has demonstrated competitive performance in network alignment.
>
> There are many papers, other than GWL or S-GWL. For example:
>
> ERIC: https://openreview.net/forum?id=lblv6NGI7un
>
> Deep Graph Matching Consensus: https://arxiv.org/abs/2001.09621
>
> Also, the complexity of S-GWL is O((V+E)K) and the current paper is O(V^3), right? In this light, I could not parse the results in Appendix A.7
>
> In a nutshell, I think this is an interesting work, but it does not add too much value to the literature--- as it is, it is a numerical optimization to QAP, which is quite rich in literature. As presented in this paper, the contribution with respect to such broad literature is quite limited. In fact, I think it may be of interest to data mining community e.g., SDM or ICDM, with significant modifications. But, I don't think this has crossed the bar of Neurips *as of now*.

---

> > ### Author Response · Authors · 2024-08-13
> >
> > Dear Reviewer T3ao,
> >
> > **QAP Vs. Network Alignment:** QAP solvers approximate the optimal solution. This approximation is of low quality unless aided with regularizers and features specific to network alignment, which we do. We have compared with generic QAP solvers and outperform them comprehensively. Our novelty lies in designing the regularizer and optimization strategies that work in the context of network alignment. Herein lies our core novelty. Furthermore, **we have outperformed well-established state-of-the-art network alignment algorithms such as S-GWL, both in accuracy and efficiency.** This further substantiates the impact of our work.
> >
> > **Supervised Network Alignment:** The referred papers for supervised learning are **inaccurate.** As we mentioned in our rebuttal, we would appreciate examples of peer-reviewed works that perform network alignment on unseen networks without retraining. In this context, we point out why the referred papers do not serve this role.
> >
> > 1. ERIC is tailored for computing graph edit distance and can't scale for large graphs. The average size of the graphs ERIC aligned is less than 30 whereas we are aligning graphs with more than 1000 nodes. Moreover, ERIC needs to be trained on every dataset separately as GNN is a function of number of features and these vary across datasets.
> >
> > 2. Same is the case with Deep graph matching consensus as it needs to be trained for every network separately.
> >
> > 3. SGWL, GWL are unsupervised and need to learn embeddings for every graph pair separately. We have already compared with these methods and comprehensively outperform them.
> >
> > **Efficiency of S-GWL:** Our empirical study includes comprehensive efficiency comparison with S-GWL and reveals its inability to scale to large datasets, which is consistent with several other works in the literature [1,2,3,4]. One contributing factor to this limitation is that S-GWL performs recursive -partitioning, a process that is not conducive to parallelization on modern CPUs with high levels of hyper-threading [4].
> >
> > [1] Zeng, Z., Zhang, S., Xia, Y., and Tong, H. Parrot: Position-aware regularized optimal transport for network alignment. WWW ’23, pp. 372–382.
> >
> > [2] Hermanns, J., Skitsas, K., Tsitsulin, A., Munkhoeva, M., Kyster, A., Nielsen, S., Bronstein, A. M., Mottin, D., and Karras, P. Grasp: Scalable graph alignment by spectral corresponding functions. ACM Trans. Knowl. Discov. Data, 17(4), Feb 2023.
> >
> > [3] Skitsas, K., Orlowski, K., Hermanns, J., Mottin, D., and Karras, P. Comprehensive evaluation of algorithms for unrestricted graph alignment. In EDBT 2023, Ioannina, Greece, March 28-31, 2023, pp. 260–272.
> >
> > [4] Li, J., Tang, J., Kong, L., Liu, H., Li, J., So, A. M.-C., and Blanchet, J. A convergent single-loop algorithm for relaxation of gromov-wasserstein in graph data. In ICLR, 2023.

---

> > > ### Author Response · Authors · 2024-08-14
> > > **Further evidence on QAP efficacy for network alignment.**
> > >
> > > Dear Reviewer T3ao,
> > >
> > > We wish to further highlight that the paper (2) you refer to in your last comment, that is the quoted line below, is the FAQ paper. FAQ is one of the primary baselines we compare to and show that it is significantly inferior to FUGAL (Figs. 1-4, Table 2).
> > >
> > > > A key application of QAP is network alignment. See for example: (1) https://arxiv.org/abs/1908.00265 (2) https://journals.plos.org/plosone/article?id=10.1371/journal.pone.0121002
> > >
> > > This serves as further evidence of:
> > >
> > > 1. Why the stated claim of generic solvers being effective on network alignment **stands on unsubstantiated grounds.**
> > > 2. The impact and novelty of the regularizers, features and optimization strategies we design for the network alignment problem that enables us to **outperform FAQ and 12 other baselines** in a comprehensive benchmarking exercise.
> > >
> > > Our extensive benchmarking exercise provides compelling evidence of the inefficacy of generic QAP solvers for network alignment, thereby underscoring the significance and contributions of our work.
> > >
> > > We hope this clarification helps to highlight the merits of our approach in the context of existing solutions for network alignment.
> > >
> > > Thank you for your consideration.
> > >
> > > Best regards,
> > >
> > > The Authors

---

### Author Rebuttal · Authors · 2024-08-05

We sincerely thank the reviewers for their insightful and constructive feedback. Below, we provide a comprehensive point-by-point response to their comments. Additionally, we attach a PDF document containing plots of several new empirical analyses as suggested by the reviewers. The key revisions and insights include:

1. **Enhanced Empirical Benchmarking:**
   * Integration of two new QAP-based baselines: Path [1] and DSPP [2]. (`Fig. 1 in PDF`)
   * Addition of Frobenius Norm and Matched Neighborhood Consistency performance measures. (`Fig. 2 in PDF`)
   * Enhanced ablation study highlighting the importance of each feature in the LAP regularizer and the impact of $\lambda$ in guiding the doubly-stochastic matrix towards a quasi-permutation matrix. (`Fig. 3 in PDF`)

2. **Expanded Related Works:**
   * We will cite and discuss various related works pointed out by Reviewer `ts95`. Two of these works [1,2] have been incorporated as new baselines.

3. **Clarifications and Improvements:**
   * We provided will better highlight the theoretical characterization of our work on algorithm convergence and termination.
   * Better consolidation of limitations and future scope for improvement.
   * Incorporation of standard deviation and other plot enhancements as suggested by Reviewer `b2PL`.

We believe these revisions significantly strengthen our manuscript. We are open to further engagement with the reviewers for any additional queries or suggestions. In light of these improvements, we kindly request the reviewers to reassess their ratings of our work.


[1] Zaslavskiy, M., Bach, F., & Vert, J. P. (2008). A path following algorithm for the graph matching problem. IEEE Transactions on Pattern Analysis and Machine Intelligence, 31(12), 2227-2242.

[2] Nadav Dym, Haggai Maron, and Yaron Lipman. 2017. DS++: a flexible, scalable and provably tight relaxation for matching problems. ACM Trans. Graph. 36, 6, Article 184 (December 2017).

---

> ### Comment · Senior_Area_Chairs · 2024-08-12
> **Please address author responses**
>
> Dear Reviewers,
>
> The paper has received a wide range of reviews.
>
> Please take a look at the other reviewer comments as well as the author feedback, and provide a response: at a minimum, acknowledging that you have looked at the responses; and adjusting your review and/or rating if you feel it is appropriate.
>
> Please do this ASAP.
>
> Thank you,
>
> SAC

---

### Decision · Program_Chairs · 2024-09-25

**Decision:**

Accept (poster)

**Comment:**

The paper considers the problem of graph alignment, where the objective is to find an optimal alignment between two graphs, and it attempts to solve the problem wht a QAP and augmenting it with a LAP regularizer for tractability. The QAP finds a permutation matrix directly on the adjacency matrices of both the graphs and the augmented LAP uses structural features of the nodes to enhance node similarity of the matchings.  Reviewers saw novelty in the method, and good evaluation.  One outlying reviewer seemed to ask for clarifications and improvements that were not the main focus of the paper.